# A population of adult satellite-like cells in *Drosophila* is maintained through a switch in RNA-isoforms

**Hadi Boukhatmi, Sarah Bray***

Department of Physiology, Development and Neuroscience, University of Cambridge, Cambridge, United Kingdom

**Abstract** Adult stem cells are important for tissue maintenance and repair. One key question is how such cells are specified and then protected from differentiation for a prolonged period. Investigating the maintenance of *Drosophila* muscle progenitors (MPs) we demonstrate that it involves a switch in *zfh1/ZEB1* RNA-isoforms. Differentiation into functional muscles is accompanied by expression of *miR-8/miR-200*, which targets the major *zfh1-long* RNA isoform and decreases Zfh1 protein. Through activity of the Notch pathway, a subset of MPs produce an alternate *zfh1-short* isoform, which lacks the *miR-8* seed site. Zfh1 protein is thus maintained in these cells, enabling them to escape differentiation and persist as MPs in the adult. There, like mammalian satellite cells, they contribute to muscle homeostasis. Such preferential regulation of a specific RNA isoform, with differential sensitivity to miRs, is a powerful mechanism for maintaining a population of poised progenitors and may be of widespread significance.

DOI: https://doi.org/10.7554/eLife.35954.001

## Introduction

Growth and regeneration of adult tissues depends on stem cells, which remain undifferentiated while retaining the potential to generate differentiated progeny. For example, muscle satellite cells (SCs) are a self-renewing population that provides the myogenic cells responsible for postnatal muscle growth and muscle repair (*Chang and Rudnicki, 2014*). One key question is how tissue specific stem cells, such as satellite cells, are able to escape from differentiation and remain undifferentiated during development, to retain their stem cell programme though-out the lifetime of the animal.

It has been argued that the progenitors of *Drosophila* adult muscles share similarities with satellite cells and thus provide a valuable model to investigate mechanisms that maintain stem cell capabilities (*Aradhya et al., 2015*; *Figeac et al., 2007*). After their specification during embryogenesis, these muscle progenitors (MPs) remain undifferentiated throughout larval life before differentiating during pupal stages. For example, one population of MPs is associated with the wing imaginal disc, which acts as a transient niche, and will ultimately contribute to the adult flight muscles. These MPs initially divide symmetrically to amplify the population. They then enter an asymmetric division mode in which they self-renew and generate large numbers of myoblasts that go on to form the adult muscles (*Gunage et al., 2014*). In common with vertebrates, activity of Notch pathway is important to maintain the MPs in an undifferentiated state (*Gunage et al., 2014*; *Mourikis and Tajbakhsh, 2014*; *Mourikis et al., 2012*; *Bernard et al., 2010*). To subsequently trigger the muscle differentiation program, levels of Myocyte Enhancer factor 2 (Mef2) are increased and Notch signalling is terminated (*Elgar et al., 2008*; *Bernard et al., 2006*). Until now it was thought that all MPs followed the same fate, differentiating into functional muscles. However, it now appears that a subset persist into adulthood forming a population of satellite-like cells (*Chaturvedi et al., 2017* and see below)). This

*For correspondence:
sjb32@cam.ac.uk

Competing interests: The authors declare that no competing interests exist.

implies a mechanism that enables these cells to escape from differentiation, so that they retain their progenitor-cell properties.

The *Drosophila* homologue of ZEB1/ZEB2, Zfh1 (zinc-finger homeodoman 1), is a candidate for regulating the MPs because this family of transcription factors is known to repress Mef2, to counter-act the myogenic program (*Siles et al., 2013*; *Postigo et al., 1999*). Furthermore, *zfh1* is expressed in the MPs when they are specified in the embryo and was shown to be up-regulated by Notch activity in an MP-like cell line (DmD8) (*Figeac et al., 2010*; *Krejcí et al., 2009*). In addition, an important regulatory link has been established whereby microRNAs (miRs) are responsible for down-regulating ZEB/Zfh1 protein expression to promote differentiation or prevent metastasis in certain contexts (*Zaravinos, 2015*; *Vandewalle et al., 2009*). For example, the miR-200 family is significantly up-reg-ulated during type II cell differentiation in fetal lungs, where it antagonizes ZEB1 (*Benlhabib et al., 2015*). Likewise, *miR-8*, a *miR-200* relative, promotes timely terminal differentiation in progeny of Drosophila intestinal stem cells by antagonizing *zfh1* and *escargot* (*Antonello et al., 2015*). Con-versely, down-regulation of *miR-200* drives epithelial mesenchymal transition (EMT) to promote metastasis in multiple epithelial derived tumours (*Korpal et al., 2008*; *Park et al., 2008*). Such observations have led to the proposal that the ZEB/miR-200 regulatory loop may be important in the maintenance of stemness, although examples are primarily limited to cancer contexts and others argue that the primary role is in regulating EMT (*Antonello et al., 2015*; *Brabletz and Brabletz, 2010*). The MPs are thus an interesting system to investigate whether this regulatory loop is a gate-keeper for the stem cell commitment to differentiation.

To investigate the concept that ZEB1/Zfh1 could be important in sustaining progenitor-type sta-tus, we examined the role and regulation of *zfh1* in *Drosophila* MPs/SCs. Our results show that *zfh1* plays a central role in the maintenance of undifferentiated MPs and, importantly, that is expression is sustained in a population of progenitors that persist in adults (pMPs) through the activity of Notch. Specifically these pMPs express an alternate short RNA isoform of *zfh1* that cannot be targeted by *miR-8*. In contrast, the majority of larval precursors express a long isoform of *zfh1*, which is subject to regulation by *miR-8* so that Zfh-1 protein levels are suppressed to enable differentiation of myo-cytes. Expression of alternate *zfh1-short* isoform is thus a critical part of the regulatory switch to maintain a pool of progenitor 'satellite-like' cells in the adult. This type of regulatory logic, utilizing RNA isoforms with differential sensitivity to miRs, may be of widespread relevance for adult stem cell maintenance in other tissues.

## Results

### Zfh1 is required for maintenance of muscle progenitors

*As zfh1* was previously shown to antagonize myogenesis (*Siles et al., 2013*; *Postigo et al., 1999*) it is a plausible candidate to maintain the muscle progenitor (MP) cells in *Drosophila* and prevent their differentiation. Its expression is consistent with this hypothesis as Zfh1 is present throughout the large group of MPs associated with the wing disc, which can be distinguished by the expression of Cut (Ct) (*Figure 1A–A''*). At early stages Zfh1 expression is uniform (*Figure 1—figure supplement 1*), but at later stages the levels become reduced in the cells with high Cut expression (*Figure 1A''*). These cells give rise to the direct flight muscles (DFMs), whereas the remaining MPs, where Zfh1 expression is high, give rise to the indirect flight muscles (IFMs) (*Figure 1A''*; *Sudarsan et al., 2001*). Zfh1 expression in MPs is therefore regulated in a manner that correlates with different differentia-tion programs.

To determine whether Zfh1 is required in the MPs to antagonize myogenic differentiation we tested the consequences from silencing *zfh1* specifically in MPs, using *1151-Gal4* to drive expression of interfering RNAs (RNAi). Two independent RNAi lines led to the premature expression of Tropo-myosin (Tm), a protein normally expressed in differentiated muscles, in the most severe (KK103205 line)~80% of *zfh1*-depleted wing discs exhibited Tm expression (*Figure 1B–C* and *Figure 1—figure supplement 1D–G*). Similarly, expression of a Myosin Heavy Chain (MHC) reporter was detected in ~20% of *zfh1* (KK103205) depleted wing discs (*Figure 1—figure supplement 1H–I*) indicating that small muscle fibers had formed precociously. Consistent with the premature expression of these muscle differentiation markers, decreased *zfh1* led to abnormal β−3Tubulin staining, showing that the residual cells had altered cell morphology in 90% of *zfh1*-depleted wing discs, (*Figure 1B'–C'*).

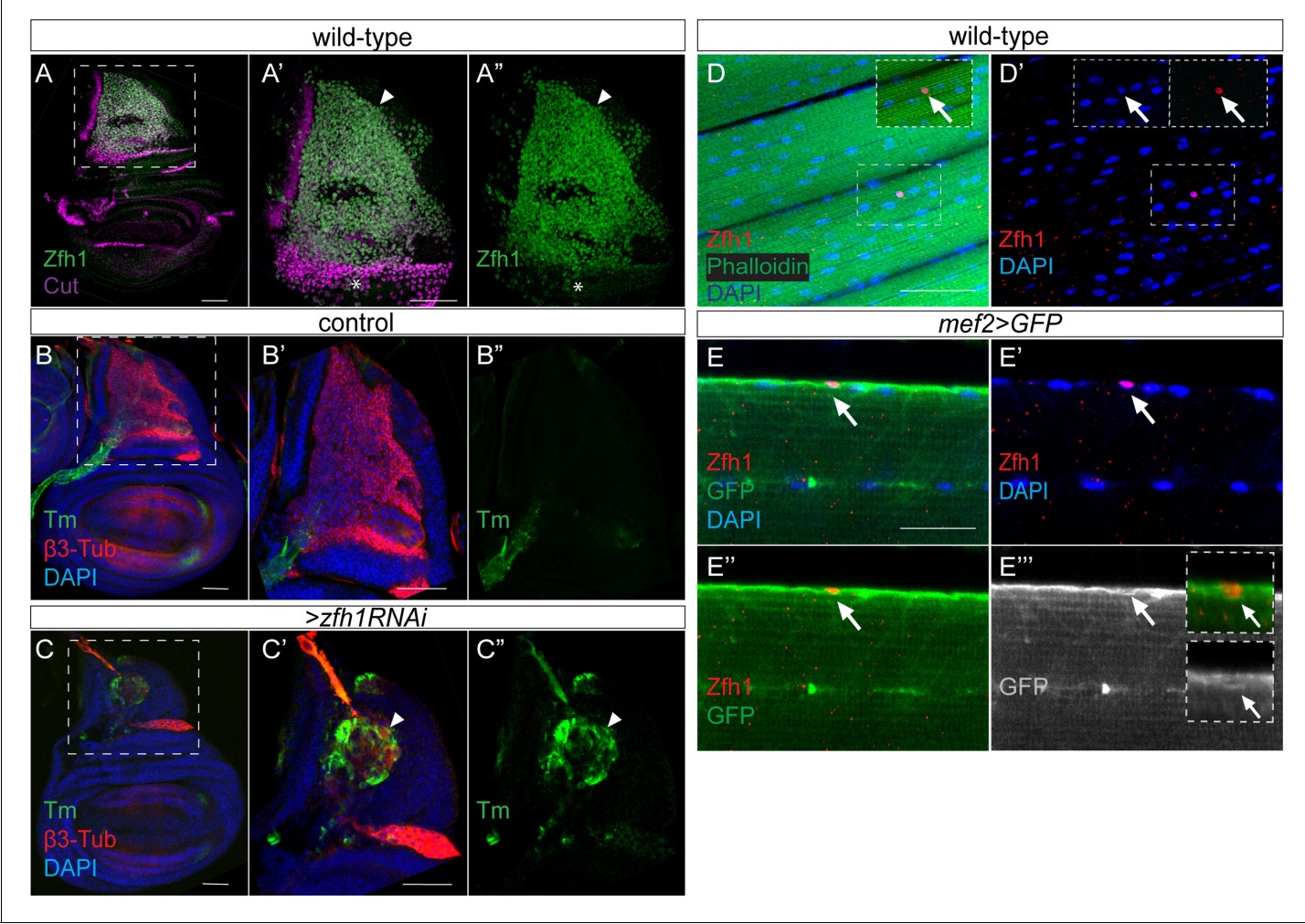

**Figure 1.** Zfh1 expression and function in MPs and in adult pMPs. (A–A") Zfh1 (Green) and Cut (Purple) expression in MPs associated with third instar wing discs, (A'–A") higher magnification (3X) of boxed region in A. Zfh1 is present in all MPs, but those with highest Cut expression have lower levels of Zfh1 (asterisk). Scale bars: 50 µM, (n > 30 wing discs from three biological replicates). (B–C) Down regulation of *zfh1* induces premature differentiation of the MPs (arrowhead in C'-C"). β3-Tubulin (β3-Tub, Red) and Tropomyosin (Tm, Green) expression in control (B, *1151-Gal4 > wRNAi*) and Zfh1 depleted (C, *1151-Gal4 > zfh1* RNAi) third instar wing discs, (B'–C") higher magnification (3X) of boxed regions in B and C. (n > 20 wing discs; from three biological replicates). (D–D') Zfh1 expression (red) indicates the existence of persistent muscle progenitors (pMPs; arrows) associated with the muscle fibres (Phalloidin (Green), DNA/Nuclei (Blue); n > 10 heminota; from three biological replicates). The immune cell marker P1 was included in the immunostaining and is absent from the pMPs (see *Figure 1—figure supplement 2*). Scale bars: 50 µM. (E–E") Zfh1 (Red) expressing pMPs (e.g. arrows in E''') are closely embedded in the muscle lamina of the adult indirect flight muscles and express Mef2 (myogenic cells; *Mef2-Gal4 >Src::GFP, green*). Nuclei (Blue), Scale bars: 25 µM, (n > 10 heminota; from two biological replicates).

DOI: https://doi.org/10.7554/eLife.35954.002

The following figure supplements are available for figure 1:

**Figure supplement 1.** Zfh1 expression and down regulation in MPs.
DOI: https://doi.org/10.7554/eLife.35954.003

**Figure supplement 2.** A population of plasmatocytes associated with the adult flight muscles expresses Zfh1 but not Mef2.
DOI: https://doi.org/10.7554/eLife.35954.004

These results demonstrate that reduced *zfh1* expression causes MPs to initiate the muscle differentiation program indicating that Zfh1 is required to prevent MP differentiation.

Lineage tracing experiments suggest that a subset of wing disc MPs have characteristics of muscle stem cells and remain undifferentiated even in adult *Drosophila*. Recently, these have been shown to express Zfh1 (*Chaturvedi et al., 2017*; *Gunage et al., 2014*), which is compatible with our observation that Zfh1 is necessary to prevent differentiation in MPs. In agreement, adult IFM muscle

fibers were associated with sparse nuclei that retained high levels of Zfh1 expression whereas the differentiated muscle nuclei exhibited no detectable expression (*Figure 1D–D'*). To better characterise these Zfh1 positive (+ve) adult cells, we expressed a membrane-tagged GFP (*UAS-Src::GFP*) under the control of a specific muscle driver *Mef2-Gal4* (*Mef2 >GFP*). This confirmed that Zfh1 was expressed in myogenic *Mef2 >GFP* expressing cells, and revealed that these cells were closely embedded in the muscle lamina (*Figure 1E–E'''*). Although many of the Zfh1 expressing cells were clearly co-expressing *mef2 >GFP*, Zfh1 was also detected in another population that lacked Mef2 expression. Often clustered, these cells were co-labelled with a plasmatocyte marker P1/Nimrod indicating that they are phagocytic immune cells (*Figure 1—figure supplement 2*). A subset of the Zfh1 +ve cells are therefore myogenic and have characteristics of persistent muscle progenitors that likely correspond to the so-called adult satellite cells recently identified by others (*Chaturvedi et al., 2017*) (*Figure 1D–E*).

The results demonstrate that Zfh1 is expressed in MPs, where it is required for their maintenance, and that its expression continues into adult-hood in a small subset of myogenic cells. If, as these data suggest, Zfh1 is important for sustaining a population of a persistent adult progenitors, there must be a mechanism that maintains Zfh1 expression in these cells while the remainder differentiate into functional flight muscles.

## *zfh1* enhancers conferring expression in MPs

To investigate whether the maintenance of Zfh1 expression in larval and adult MPs could be attributed to distinct enhancers, we screened *enhancer-Gal4* collections (*Jenett et al., 2012*; *Jory et al., 2012*; *Manning et al., 2012*) to identify *zfh1* enhancers that were active in larval MPs. From the fifteen enhancers across the *zfh1* genomic locus that were tested, (*Figure 2* and *Figure 2—figure supplement 1A*) three directed GFP expression in the Cut expressing MPs at larval stages (*Figure 2B–D*). These all correlated with regions bound by the myogenic factor Twist in MP-related cells (*Figure 2—figure supplement 1A*; *Bernard et al., 2010*). Enhancer 1 (*Enh1*; VT050105) conferred weak expression in scattered progenitors (*Figure 2B*). Enhancer 2 (*Enh2*; VT050115) was uniformly active in all MPs and also showed ectopic expression in some non-Cut expressing cells (*Figure 2C*). Finally, Enhancer 3 (*Enh3*; GMR35H09) conferred expression in several MPs with highest levels in a subset located in the posterior (*Figure 2D*). *Enh3* encompasses a region that was previously shown to be bound by Su(H) in muscle progenitor related cells, hence may be regulated by Notch activity (*Figure 2—figure supplement 1A*; *Bernard et al., 2010*; *Krejcí et al., 2009*). These results demonstrate that several enhancers contribute to *zfh1* expression in the MPs.

To determine which enhancer(s) are also capable of conferring *zfh-1* expression in adult pMPs we assessed their activity in adult muscle preparations. Only *Enh3* exhibited any activity in these cells (*Figure 2E*), where it recapitulated well Zfh1 protein expression (*Figure 2E–E'*). Thus, *Enh3-GFP* was clearly detectable in scattered cells, which were closely apposed to the muscle fibers and contained low levels of Mef-2 (*Figure 2E'''*), and was not expressed in differentiated muscle nuclei (*Figure 2*).

During pupal stages MPs migrate and surround a set of persistent larval muscles that act as scaffolds for the developing IFMs (*Roy and VijayRaghavan, 1998*; *Fernandes et al., 1991*). By 18–22 hr after puparium formation (APF), fusion of myoblasts is ongoing and by 30–36 hr APF, most myoblasts have fused and myogenesis is advanced. By this stage, Zfh1 expression is already restricted to single cells (*Chaturvedi et al., 2017*). We therefore examined *Enh3* activity during the pupal period, using *Enh3-Gal4 > UAS* GFP, which yields a higher level of expression than the direct *Enh3-GFP* fusion. At 18–22 hr APF, *Enh3* activity was detected in both differentiating myoblasts (inside muscle templates) and undifferentiated MPs (between and around muscle templates) (*Figure 2F'–F''*). Importantly, a subset of MPs located between muscle templates exhibited higher levels of *Enh3* expression and of Zfh1 levels (*Figure 2F'*), whereas lower levels were present in the differentiating myoblasts. By 30–36 hr APF *Enh3* expression was restricted to pMPs, which expressed high level of Zfh1 and were closely apposed to the muscle fibers (*Figure 2G–G''*). At the same stage, we consistently detected a small number of Zfh1 +ve cells that lack detectable *Enh3* expression and we speculate that these are undergoing differentiation, since myogenesis is still ongoing at this stage (*Figure 2G*). In general however, there is a strong correlation between *Enh3* expression and the establishment of the adult Zfh1 +ve pMPs, suggesting that *Enh3* is responsible for maintaining *zfh1*

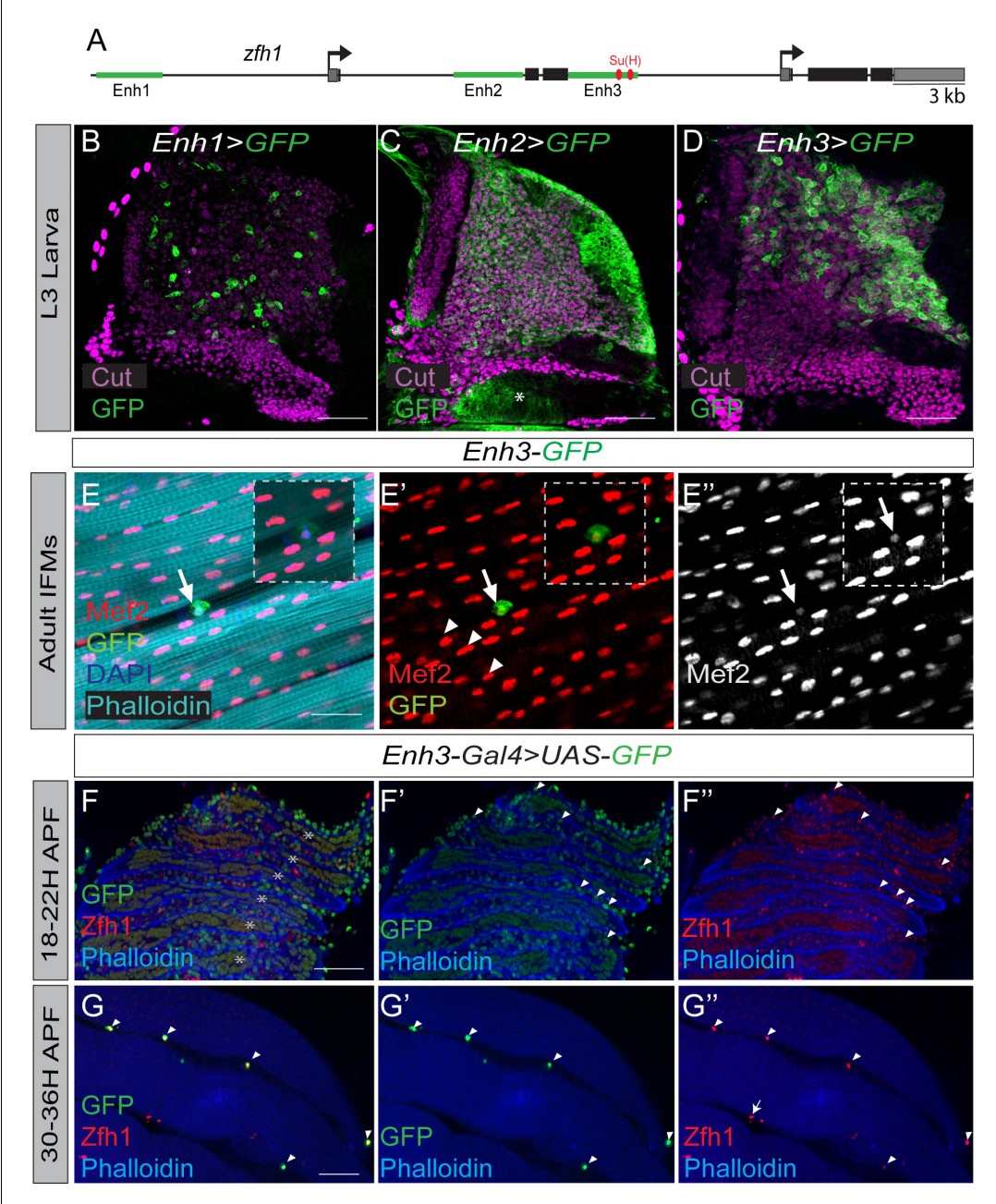

**Figure 2.** Regulation of *zfh1* in MPs and adult pMPs. (**A**) Schematic view of *zfh1* genomic region, *zfh1* regulatory enhancers are represented by green rectangles and arrows indicate transcription start-sites. Coding exons and untranslated regions are represented in black and grey boxes, respectively. (**B-D**) Three different *zfh1* enhancers are active in the MPs (labelled with Cut, purple). *Enh1* (VT050105, **B**) drives GFP (Green) in a subset of scattered MPs; *Enh2* (VT050115, **C**) drives GFP throughout the MPs, and in some non-MP cells (asterisk); *Enh3* (GMR35H09, **D**) is highly expressed in a subset of MPs located in the posterior region of the notum. Scale bars: 50 μM. (n = 30 wing discs). (**E-E''**) *Enh3-GFP* (Green) expression is maintained in adult pMPs (characterised by low levels of Mef2, red; arrows E'-E'') but not in differentiated muscle nuclei (high Mef2, red; arrowheads G'-G''). Phalloidin marks muscles (Cyan) and DAPI labels all nuclei (Blue). Insets: boxed regions magnified 12.5 X. Scale bars 25 μM. (n = 10 heminota; from two biological replicates). (**F-G**) Muscle (IFM) preparation isolated from *Enh3-Gal4 > UAS* GFP pupae at 18–22 hr APF (**F**) or at 30–36 hr APF (**G**). *Enh3-GFP* (Green) and Zfh1 (Red) are detected in MPs. (**F**) At 18–22 hr *Enh3-GFP* activity is higher (arrowheads in E') in some undifferentiated MPs located between muscle templates (muscles are labeled with Phalloidin, Blue, asterisks). (**G**) At 30–36 hr APF, *Enh3-GFP* (Green) activity and Zfh1 (Red) are detected in pMPs (arrowheads) and not in differentiated muscle nuclei. A few Zfh1 +ve cells do not express *Enh3-GFP* (Arrows G''). Note: anti-P1, an immune cell marker, was included in the staining to exclude plasmatocytes from the analysis (see **Figure 1—figure supplement 2**).

DOI: https://doi.org/10.7554/eLife.35954.005

The following figure supplement is available for figure 2:

*Figure 2 continued on next page*

*Figure 2 continued*

**Figure supplement 1.** Identification of *zfh1* enhancers active in MPs and pMPs.

DOI: https://doi.org/10.7554/eLife.35954.006

transcription in this progenitor population during the transitionary phase between 20 hr and 30 hr APF.

If *Enh3* is indeed responsible for expression of *zfh1* in MPs and pMPs, its removal should curtail *zfh1* expression in those cells. To test this, *Enh3* was deleted by Crispr/Cas9 genome editing (Δ*Enh3*; see Materials and methods). Δ*Enh3* homozygous flies survived until early pupal stages allowing us to analyze the phenotype at larval stages. As predicted, Δ*Enh3* MPs exhibited greatly reduced Zfh1 protein expression (*Figure 2—figure supplement 1B–D*) that correlated with decreased *zfh1* mRNA levels (*Figure 2—figure supplement 1E*). Although striking, the effects of Δ*Enh3* did not phenocopy those of depleting *zfh1* using RNAi, as no premature up-regulation of muscle differentiation markers (MHC, Tm) occurred in Δ*Enh3* discs (data not shown). This is likely due to residual *zfh1* mRNA/protein (*Figure 2—figure supplement 1*), brought about by the activity of other *zfh1* enhancers (e.g. *Enh1* and *Enh2*, *Figure 2B–C*). Nevertheless, it is evident that *Enh3* has a key role in directing *zfh1* expression in MPs/pMPs.

## Adult Zfh1 +ve MP cells contribute to flight muscles

By recapitulating Zfh1 in adult MPs, *Enh3* provides a powerful tool to investigate whether the persistent MPs are analogous to muscle satellite cells, which are able to divide and produce committed post-mitotic myogenic cells that participate in muscle growth and regeneration. To address this we used a genetic G-trace method, which involves two UAS reporters, an RFP reporter that directly monitors the current activity of the Gal4 and a GFP reporter that records the history of its expression to reveal the lineage (*Evans et al., 2009*). When *Enh3-Gal4* was combined with the G-trace cassette RFP expression was present in the muscle-associated pMPs, which have low Mef2 expression (*Figure 3A–A''*). Strikingly, most of the muscle nuclei expressed GFP (*Figure 3A'*) suggesting that they are derived from ancestral *Enh3* expressing cells. Furthermore, close examination of the *Enh3* driven RFP expression showed that it often persisted in two nearby muscle nuclei (*Figure 3A–A'*). This suggests that these nuclei are recent progeny of *Enh3*-expressing cells, indicating that these cells have retained the ability to divide, a characteristic of satellite cell populations (*Figure 3*). To further substantiate this conclusion, we verified that adult Zfh1 +ve cells were actively dividing cells, using the mitotic marker phosphohistone-3 (pH3) staining. Many Zfh1 +ve cells co-stained with pH3 indicating that these adult cells remain mitotically active (*Figure 3B–B''*).

If the mitotically active Zfh1 +ve cells are indeed important for muscle homeostasis, their progeny should become incorporated into the muscle fibres. We therefore quantified the proportion of muscle nuclei derived from the pMPs during the first 10 days of the adult life, by using a temperature sensitive Gal80 (*tubGal80ts*) to restrict *Enh3-Gal4* until eclosion and combining it with the *G-Trace* cassette to mark the progeny (*Figure 3C–D*). Strikingly, the conditional activation of *Enh3-Gal4* in adults resulted in GFP labeling of ~24% of muscle nuclei (*Figure 3D*) indicating a significant role of the pMPs in contributing to muscle maintenance. Indeed when we used a similar regime to deplete *zfh1* in pMPs and examined flies at ten days (*Figure 3E–I*) we found that ~ 30% of adult flies had a 'held out wing' posture (n = 93) (*Figure 3E–F*), a phenotype often associated with flight muscle defects (*Vigoreaux, 2001*). The number of nuclei per muscle (DLM4) was also significantly reduced (~20% fewer nuclei) in the aged adults when *zfh1* was specifically depleted in the pMPs (*Figure 3G–H*). Likewise, genetic ablation of pMPs (by expressing the pro-apoptotic gene *reaper*) led to a similar reduction in muscle nuclei (*Figure 3I*). No 'held out wing' phenotype or muscle defects were observed in adult flies within 24 hr of *zfh1* knock-down, indicating that the phenotypes at 10 days are due to a defect in the homeostasis of the adult flight muscles. Taken together the results argue that the adult Zfh1 +ve myoblasts cells resemble mammalian satellite cells, retaining the capacity to divide and provide progeny that maintain the adult flight muscles.

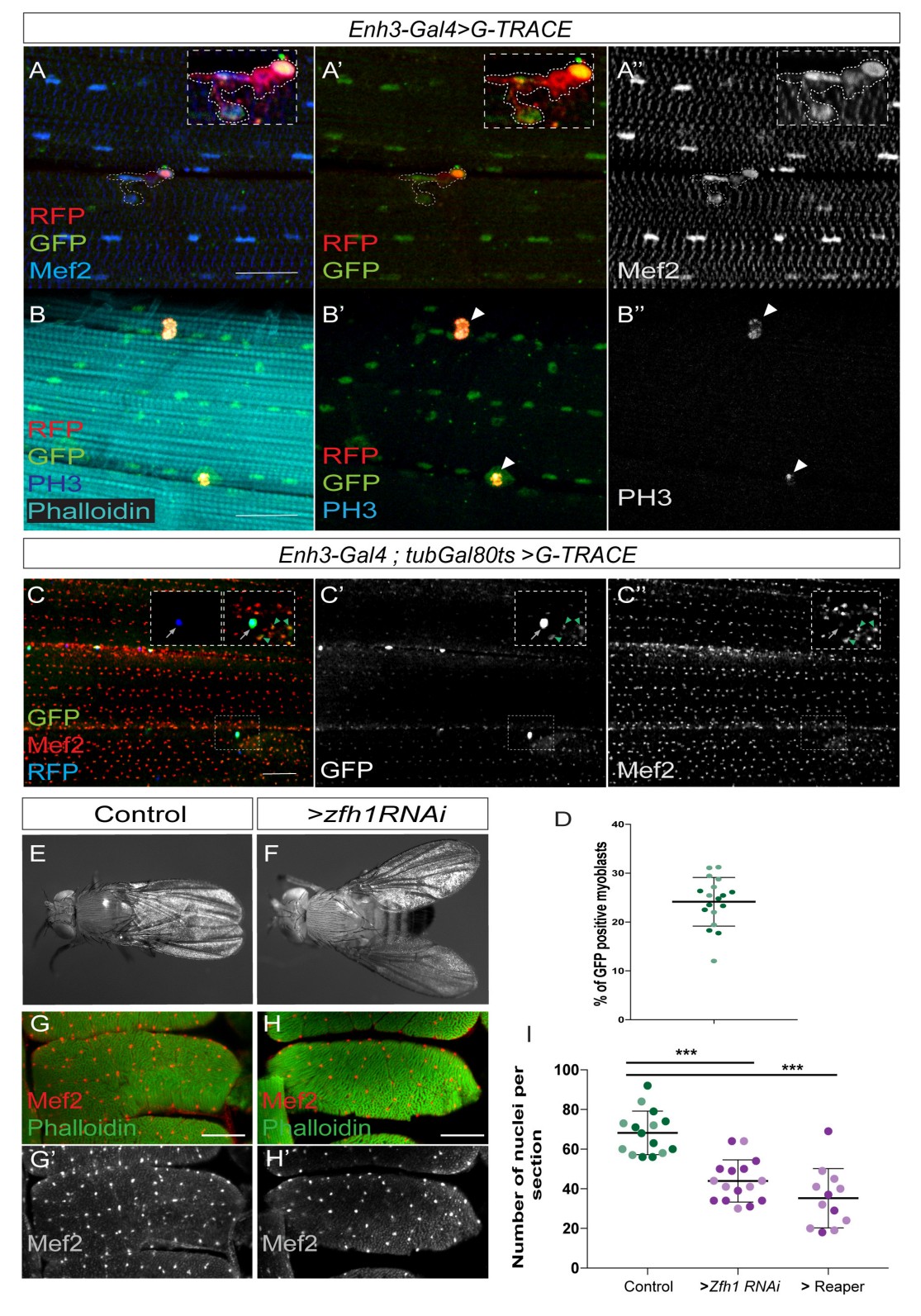

**Figure 3.** Adult pMPs contribute to muscle homeostasis. (A-A'') Lineage tracing shows that adult pMPs contribute to muscles. Cross-section of indirect flight muscles from adult flies where *Enh3-Gal4* drives expression of the *G-Trace* cassette; GFP (Green) indicates myoblasts that have expressed *Enh3-Gal4*, RFP (Red) indicates myoblasts where Gal4 is still active, Mef2 labels muscle nuclei (Blue). Note that the RFP (Red, detected with anti-RFP in A') persists in the recently born myoblasts (Mef2), which are closely localized to the pMPs. Insets: boxed regions magnified 20 X (n = 15 heminota; from
*Figure 3 continued on next page*

*Figure 3 continued*
three biological replicates). (B-B'') pMPs are mitoticaly active, indicated by anti-phosphH3 (White in B''). (PH3 detected in 47% of pMPs, n = 80; from three biological replicates). (C-C'') Cross-section of indirect flight muscles from adult flies (*Enh3-Gal4; tubGal80ts > G Trace*) where *Enh3-Gal4* directed *G-Trace* activity was induced for 10 days after animal hatching. GFP (Green, Arrowheads, C') indicates descendants of pMPs (Blue, Arrows); Mef2 labels muscle nuclei (Red, White). (D) Proportion of newly born myoblasts (marked by GFP; e.g. C') relative to total number of myoblasts (marked by Mef2; e.g. C'') in muscle preparations. (n = 18 heminota; light and dark shading indicates data points collected from two independent replicates replicates). Scale bar: 60 μM. (E-F) Prolonged *zfh1* depletion in pMPs (10 days after adult hatching) leads to a 'held out' wings posture; dorsal view of (E) control (*Enh3-Gal4; tubGal80ts > UAS wRNAi;*) and (F) *zfh1* depletion (*Enh3-Gal4; tubGal80ts > UAS-zfh1RNAi* (KK 103205)) adult flies. (G-H) Transverse sections of DLM4 muscle stained with Phalloidin (Green) and Mef2 (Red, White) from the indicated genotypes. Fewer Mef2 +ve nuclei are present in muscles when *zfh1* is depleted. Scale bars: 50 μM. (I) Similar reductions in muscle nuclei occur following *zfh1* depletion (*Enh3-Gal4; tubGal80ts > UAS-zfh1RNAi*) or following genetic ablation of pMPs, via expression of the pro-apoptotic gene *reaper* (*rpr; Enh3-Gal4;tubGal80ts > UAS* rpr). The number of nuclei per section in the indicated conditions was significantly different, light and dark shading indicates data points collected from two independent replicates (>*zfh1 RNAi* ***p=0.0013, n = 16;>*Rpr* ***p<0.0001, n = 12).
DOI: https://doi.org/10.7554/eLife.35954.007

## Notch directly regulates *zfh1* expression in muscle progenitors and adults pMPs

As mentioned above, *zfh1* is regulated by Notch activity in *Drosophila* DmD8, MP-related, cells (*Krejcí et al., 2009*), where *Enh3* was bound by Su(H) (*Figure 2A* and *Figure 2—figure supplement 1*). Furthermore, phenotypes from depletion of *zfh1* in MPs, were reminiscent of those elicited by loss of Notch (N) signaling (*Figure 1* and *Krejcí et al., 2009*). Notch activity is therefore a candidate to maintain Zfh1 expression in the adult pMPs, thereby preventing their premature differentiation. As a first step to test whether Notch activity contributes to *zfh1* expression, we depleted *Notch* in muscle progenitors by driving *Notch RNAi* expression with *1151-Gal4* (*Figure 4A–C*). Under these conditions, Zfh1 levels were significantly reduced, consistent with Notch being required for *zfh1* expression in MPs. Second, the consequences of perturbing Notch regulation by mutating the Su(H) binding motifs in *Enh3* were analyzed. Two potential Su(H) binding sites are present in *Enh3* and both are highly conserved across species (*Figure 2A*). Mutation of both motifs, *Enh3[mut]*, resulted in a dramatic decrease of the enhancer activity in the MPs (*Figure 4D–F*). This supports the hypothesis that Notch directly controls *zfh1* expression in MPs by regulating activity of *Enh3*.

Since *Enh3* activity persists in the adult pMPs (*Figure 2*), we next analyzed whether mutating the Su(H) motifs impacted expression in these adult pMPs. Similar to larval stage MPs, *Enh3[mut]* had lost the ability to direct expression of GFP in the adult pMPs (*Figure 4G–H*). Thus, the Su(H) motifs are essential for *Enh3* to be active in the adult pMPs. These data support the model that persistence of Zfh1 expression in adult MPs is likely due to Notch input, acting through *Enh3*.

The results imply that Notch should be expressed and active in the adult pMPs. To investigate this, we made use of a *Notch[NRE]-GFP* reporter line. *Notch[NRE]* is an enhancer from the *Notch* gene, and itself regulated by Notch activity, such that it is a read out both of Notch expression and of Notch activity (*Simón et al., 2014*). Robust expression of *Notch[NRE]-GFP* reporter was detected in Zfh1 +ve adult pMPs, confirming that Notch is active in these cells (*Figure 4I*) but not in the differentiated muscles. Together, the results show that *zfh1* expression in the adult pMPs requires Notch activity acting through *Enh3*.

## *zfh1* is silenced by the conserved microRNA *miR-8/miR-200* in MPs

Although transcriptional control of *zfh1* by Notch explains one aspect of its regulation, since all larval MPs express Zfh1 it remained unclear how a subset maintain this expression and escape from differentiation to give rise to adult pMPs. A candidate to confer an additional level of regulation on *zfh1* expression is the micro RNA *miR-8/miR-200,* which is important for silencing *zfh1* (and its mammalian homologue ZEB1) in several contexts. The regulatory loop between *miR-8/miR-200* and *zfh1/ZEB* has been extensively studied in both *Drosophila* and vertebrates and is mediated by a *miR-8/miR-200* seed site located in the 3' untranslated region (3'UTR) (*Antonello et al., 2015*; *Vallejo et al., 2011*; *Brabletz and Brabletz, 2010*). Moreover, *miR-8* was previously reported to be involved in flight muscle development (*Fulga et al., 2015*).

To determine whether *miR-8* could down-regulate *zfh1* in muscle progenitors to promote their differentiation into muscles, we first examined the spatiotemporal expression pattern of *miR-8* at

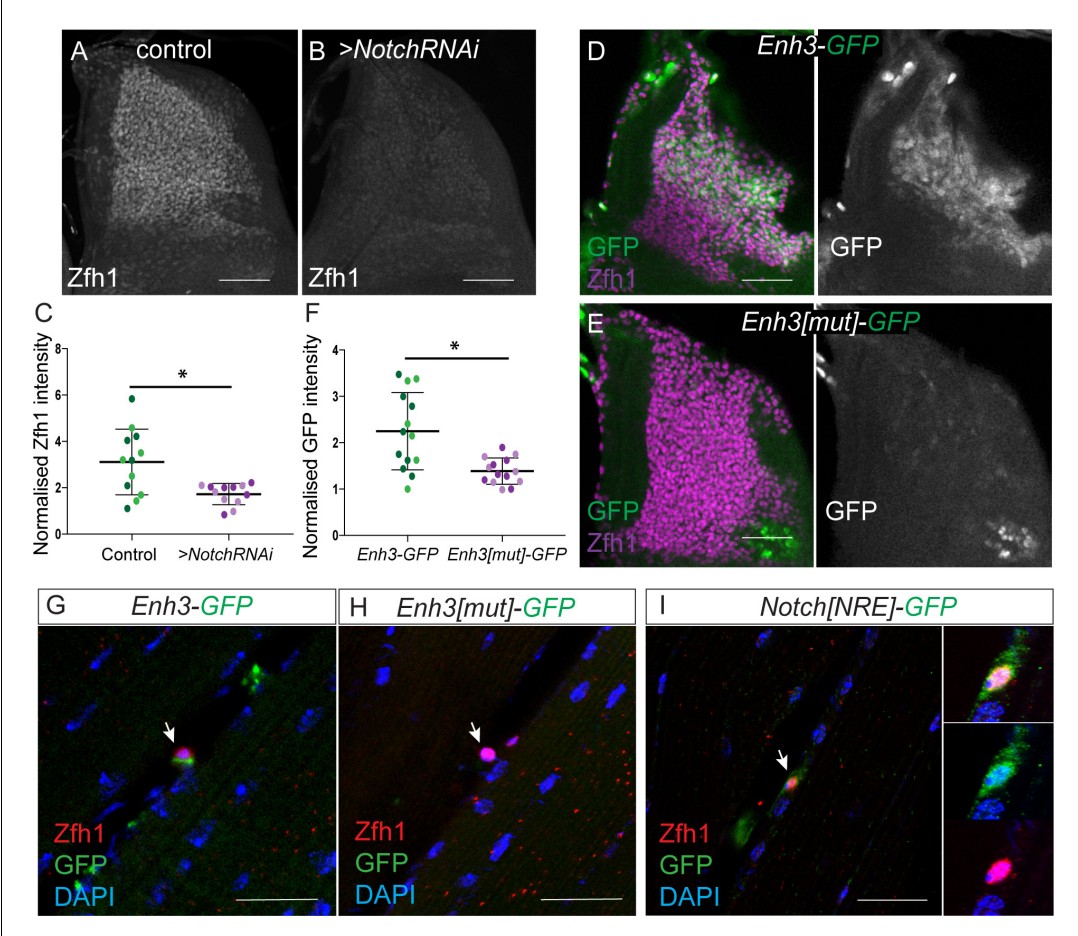

**Figure 4.** Notch directs Zfh1 expression in MPs and pMPs. (A-C) Zfh1 level (White) is significantly reduced when *Notch* is down regulated. Expression of Zfh1 in MPs (A) is severely reduced in the presence of *Notch* RNAi (B, *1151-Gal4 > UAS NotchRNAi*), Scale Bars: 50 µM. (C) Quantification of Zfh1 expression levels (*p<0.05, n = 12 wing discs in each condition, light and dark shading indicates data obtained from two independent replicates). (D-F) *Enh3* (D, *Enh3-GFP*, Green) expression in MPs (Purple, Zfh1) is abolished when Su(H) motifs are mutated (E, *Enh3[mut]-GFP*). Scale bars: 50 µM. (F) Quantification of expression from *Enh3* and *Enh3[mut]* (*p=0.022, n = 14 wing discs in each condition, light and dark shading indicates data obtained from two independent replicates). (G-H) *Enh3* (G, *Enh3-GFP*, Green) expression in adult pMPs (red, Zfh1) is abolished when Su(H) motifs are mutated (H, *Enh3[mut]-GFP*, Green), DAPI (Blue) reveals all nuclei. (I) *Notch[NRE]-GFP* (Green) is co-expressed with Zfh1 (Red) in the pMPs associated with the indirect flight muscles; DAPI (Blue) detects all nuclei. (n = 12 heminota; from two independent replicates). In G-I anti-P1 was included to label immune cells and exclude them from the analysis. Scale bars: 25 µM.

DOI: https://doi.org/10.7554/eLife.35954.008

larval, pupal and adult stages (*Figure 5*) using *miR-8-Gal4,* whose activity reflects the expression of the endogenous *miR-8* promoter (*Karres et al., 2007*). Expression of *miR-8-Gal4* was almost undetectable in the larval MPs, although it was highly expressed in the wing pouch (*Figure 5*). A low level of *miR-8-Gal4* expression was also detected in a subset of the MPs where Zfh1 levels are slightly reduced (high Ct expressing DFM precursors; *Figure 5*). Thus, expression of *miR-8* is inversely correlated with Zfh1; its overall expression is low in larval MPs where Zfh1 expression is important to prevent their differentiation (*Figure 5B–B* and *Figure 1A*).

We subsequently compared *miR8-Gal4* and Zfh1 expression in 18–22 hr APF pupae (*Figure 5D–D''*). At this stage, *miR-8-Gal4* and Zfh1 expression overlapped in most, if not all, myogenic nuclei (*Figure 5D*). However, *miR-8-Gal4* expression level was elevated in the differentiated myoblasts, which are located inside the muscle templates (*Figure 5D–D'*). Conversely, Zfh1 expression level was slightly lower in this population and higher in the undifferentiated MPs (*Figure 5D–D''*). Thus *miR-8* and Zfh1 have reciprocal low and high expression patterns in the MPs at this stage of myogenesis.

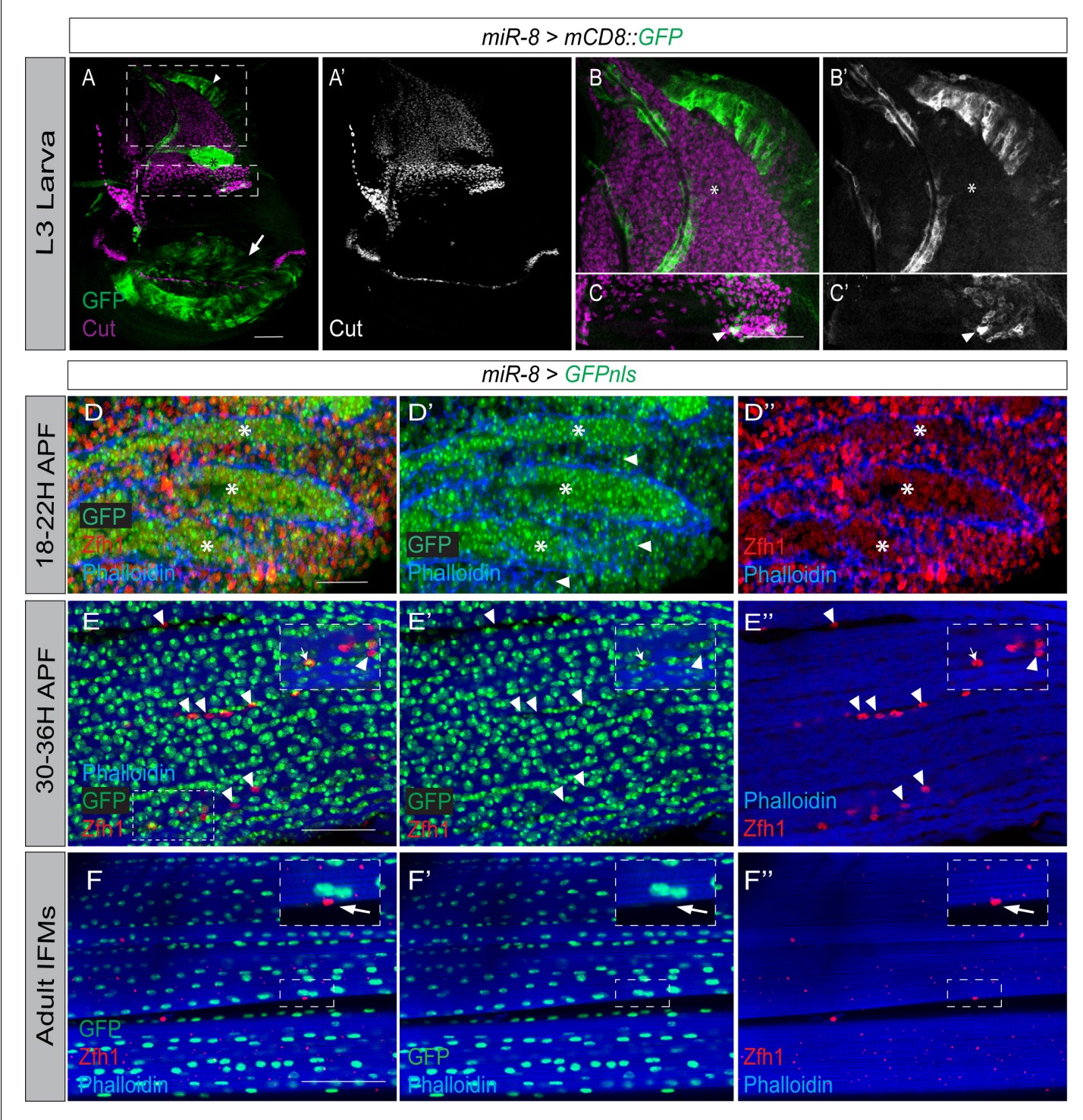

**Figure 5.** Expression dynamics of *miR-8* and Zfh1 in MPs and pMPs during indirect flight muscle development. (**A-C**) *miR-8* (Green, *miR-8-Gal4 > UAS-mCD8::GFP*) is not highly expressed in MPs (Cut, Purple) but is prevalent in the wing disc pouch (Arrow), notum (Arrowhead) and air sac (Asterisk). Higher magnification shows that low level of *miR-8* expression can be detected in the subset of MPs where Zfh1 is normally low (Arrowhead in C) but not in other MPs. Scale bars: 50 µM. (n > 20 wing discs; from three biological replicates). (**D-D''**) IFM preparation isolated from *miR-8-Gal4 > UAS-nlsGFP* pupae at 18–22 hr APF. At this stage *miR-8* (Green; **D and D'**) is co-expressed with Zfh1 (Red; **D and D''**) in MPs but *miR-8* is more highly expressed in differentiated myoblasts (Asterisks in D'), found within the muscles (labeled with Phalloidin, Blue), compared to the undifferentiated MPs, found between the muscles (Arrowheads in D'). In contrast, Zfh1 (**D, D'**) is detected at lower levels in the differentiated myoblasts compared to the undifferentiated MPs (Asterisks in D'). Scale bar: 25 µM. (**E-E''**) IFM preparation isolated from *miR-8-Gal4 > UAS-nlsGFP* pupae at 30–36 hr APF. At this

*Figure 5 continued on next page*

*Figure 5 continued*

stage, *mir-8* is highly detected in the differentiated muscle nuclei. The majority of the Zfh1 +ve pMPs (Red) do not express *mir-8* (Green) (Arrowheads). Few Zfh1 +ve pMPs express *mir-8* (Arrows). Scale bar: 50 μM. (**F-F''**) In adult IFMs, *mir-8* (Green) expression is absent from Zfh1 +ve (red) pMPs (Arrows, **D-D'**) but is present at uniformly high levels in IFMs, (Phalloidin, Blue). Scale bars: 50 μM. (n = 20 heminota; from three biological replicates).

DOI: https://doi.org/10.7554/eLife.35954.009

By 30–36 hr APF, Zfh1 expression was restricted to pMPs while *miR-8-Gal4* was predominantly expressed throughout the differentiated myoblasts (*Figure 5E*). Notably, a few of the Zfh1 +ve cells at this stage retained *miR-8-Gal4* expression (*Figure 5E–E'*). Similar to 30–36 hr APF, adult muscles had uniform and high levels of *miR-8-Gal4* (*Figure 5F–F''*). However, at this time, *miR-8-Gal4* expression was totally absent from all Zfh1 +ve adult pMPs (*Figure 5F–F''*). These data show that, during muscle formation, *miR-8* expression level is inversely correlated to Zfh1; supporting the model that *miR-8* negatively regulates *zfh1*.

Given their complementary expression patterns we next tested the impact of *miR-8* overexpression on Zfh1 protein levels in larval MPs (*Figure 6A–B*). Zfh1 protein levels were significantly diminished under these conditions, in agreement with *miR-8* regulating *zfh1* post-transcriptionally (*Figure 6C*). Although this manipulation was not sufficient to cause premature up-regulation of muscle differentiation markers or associated morphological changes in MPs, the few surviving adults all displayed a held-out wing phenotype, which is often associated with defective flight muscles (*Vigoreaux, 2001*). Next we assessed whether *miR-8* activity/expression changes in response to Mef2 levels, a critical determinant of muscle differentiation, using a *miR-8* sensor (containing two *miR-8* binding sites in its 3'UTR [*Kennell et al., 2012*]). Expression of the *miR-8* sensor was specifically decreased when Mef2 was overexpressed in MPs, suggesting that *miR-8* expression responds to high level of Mef2 (*Figure 6—figure supplement 1*).

If down-regulation of *zfh1* by *miR-8* is important to allow muscle differentiation, selective depletion of *miR-8* should allow more MPs to escape differentiation. To achieve this, a *miR-8 sponge* construct (UAS-miR-8Sp; *Fulga et al., 2015*) was expressed using *Mef2-Gal4*, so that it would decrease *miR-8* activity in differentiating myoblasts. Adult muscles were still formed under these conditions. However the final number of pMPs was significantly increased (*Figure 6D–F*). Conversely ectopic expression of *miR-8* in adult MPs (using *Enh3-Gal4*) led to a reduction in the number of muscle nuclei similar to that seen with *zfh1* down-regulation (*Figure 6—figure supplement 1D*). Together, these data argue that *miR-8* up-regulation during muscle differentiation blocks Zfh1 production to allow MPs to differentiate and if its expression is compromised, more MPs are permitted to escape differentiation. It is also possible that *miR-8* has additional targets, besides Zfh1, that are involved in maintenance/differentiation of MPs.

## An alternate short *zfh1* isoform is transcribed in adult pMPs

To retain their undifferentiated state, the adult pMPs must evade *miR-8* regulation and maintain Zfh1 expression. The *zfh1* gene gives rise to three different mRNA isoforms; two long *zfh1* isoforms (*zfh1-long*; *zfh1-RE/RB*) and one short *zfh1* isoform (*zfh1-short*; *zfh1-RA*) (*Figure 7A*). Although *zfh1-long* isoforms have two additional N-terminal zinc fingers, all three RNA-isoforms produce proteins containing the core zinc finger and homeodomains needed for Zfh1 DNA-binding activity (*Figure 7—figure supplement 1A*; *Postigo et al., 1999*). Importantly, *zfh1-short* isoform has a shorter 3'UTR, which lacks the target site for *miR-8* (*Figure 7A*; *Antonello et al., 2015*), as well as differing in its transcription start site (TSS) (*Figure 7A*; Flybase FBgn0004606). The lack of a seed site makes *zfh1-short* insensitive to *miR-8* mediated down-regulation. This means that *zfh1-short* expression would enable cells to retain high level of Zfh1 protein, even in the context of *miR-8* expression and, if present in a subset of MPs, could explaining how they can escape differentiation.

To determine whether *zfh1-short* is indeed expressed in MPs, we designed fluorescent probes specific for the *zfh1-long* and *zfh1-short* isoforms and used them for in situ hybridization (FISH) at larval and adult stages (*Figure 7A*). In larval stages (L3), *zfh1-long* isoforms were present at uniformly high levels in the MPs (*Figure 7B–B'''*) whereas *zfh1-short* was expressed at much lower levels and only detected in a few MPs in each disc (*Figure 7C–C'''*). In adult muscles, where pMPs were marked by *Enh3-Gal4 > GFP* and low Mef2 expression (*Figure 7D–E*), high levels of *zfh1-short* and much

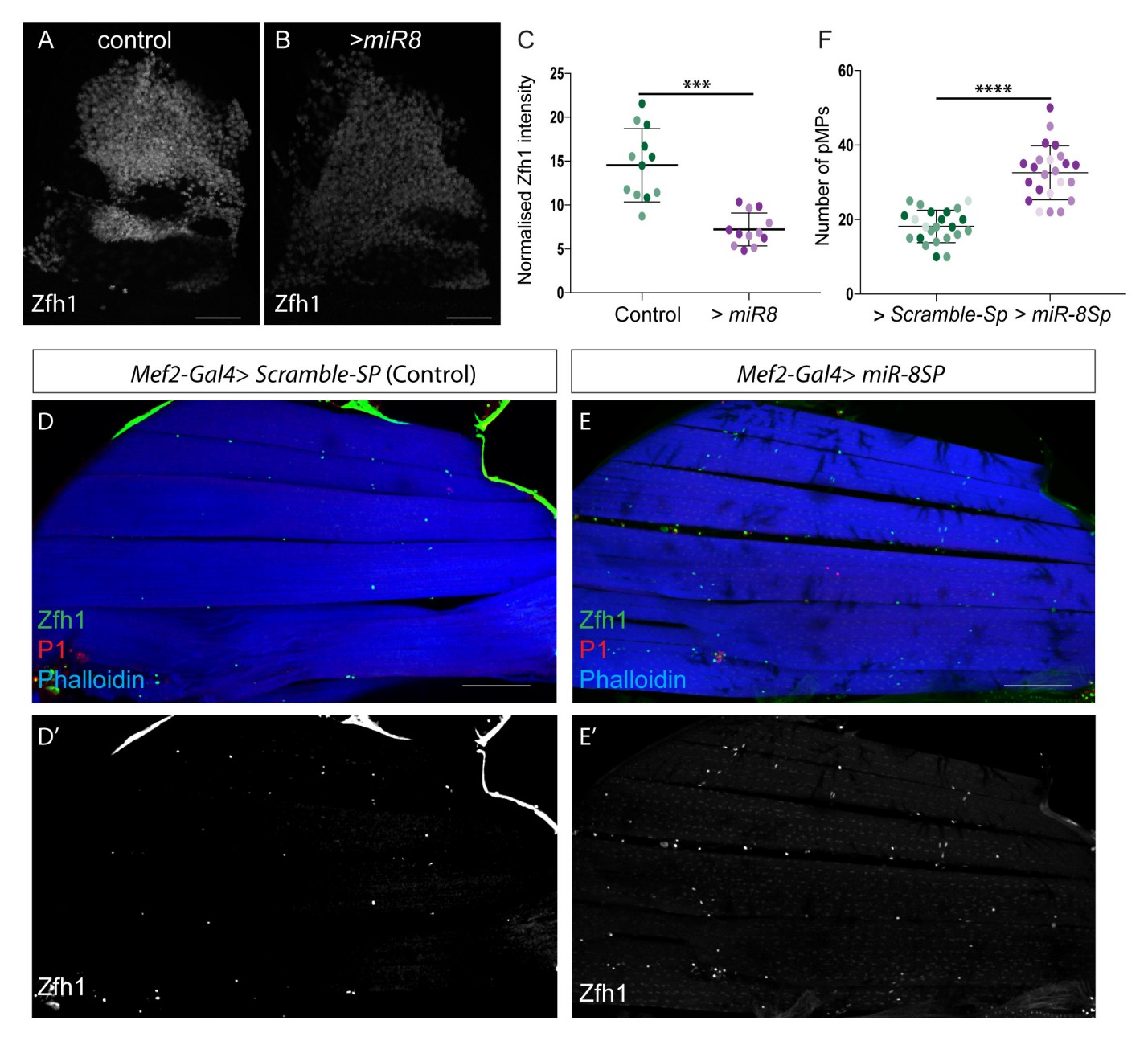

**Figure 6.** The conserved microRNA *miR-8/miR-200* antagonizes *zfh1* to promote muscle differentiation. (**A-C**) Effect of *miR-8* overexpression (*1151-Gal4 > UAS-miR-8*) on Zfh1 (White) protein level in MPs. Scale Bars: 50 μM. (**C**) Zfh1 expression is significantly reduced by *miR-8* over-expression. (***p=0.0009, n = 12 wing discs in each condition, light and dark shading indicates data points from two independent replicates). (**D-E**) Sagittal sections of adult IFMs stained for Phalloidin (Blue), Zfh1 (Green) and P1 (Red). Down regulating *miR-8* during muscle differentiation (*Mef2-Gal4 > UAS-miR-8-Sp*) increases the final number of adult pMPs. (**F**) The number of pMPs in adult IFMs in the indicated conditions was significantly different. (****p<0.0001, n = 18 adults for each genotype; light, dark and intermediate shading indicates data points from three independent replicates). Scale bars: 100 μM.
DOI: https://doi.org/10.7554/eLife.35954.010

The following figure supplement is available for figure 6:

**Figure supplement 1.** *miR-8* responds to high level of Mef2.
DOI: https://doi.org/10.7554/eLife.35954.011

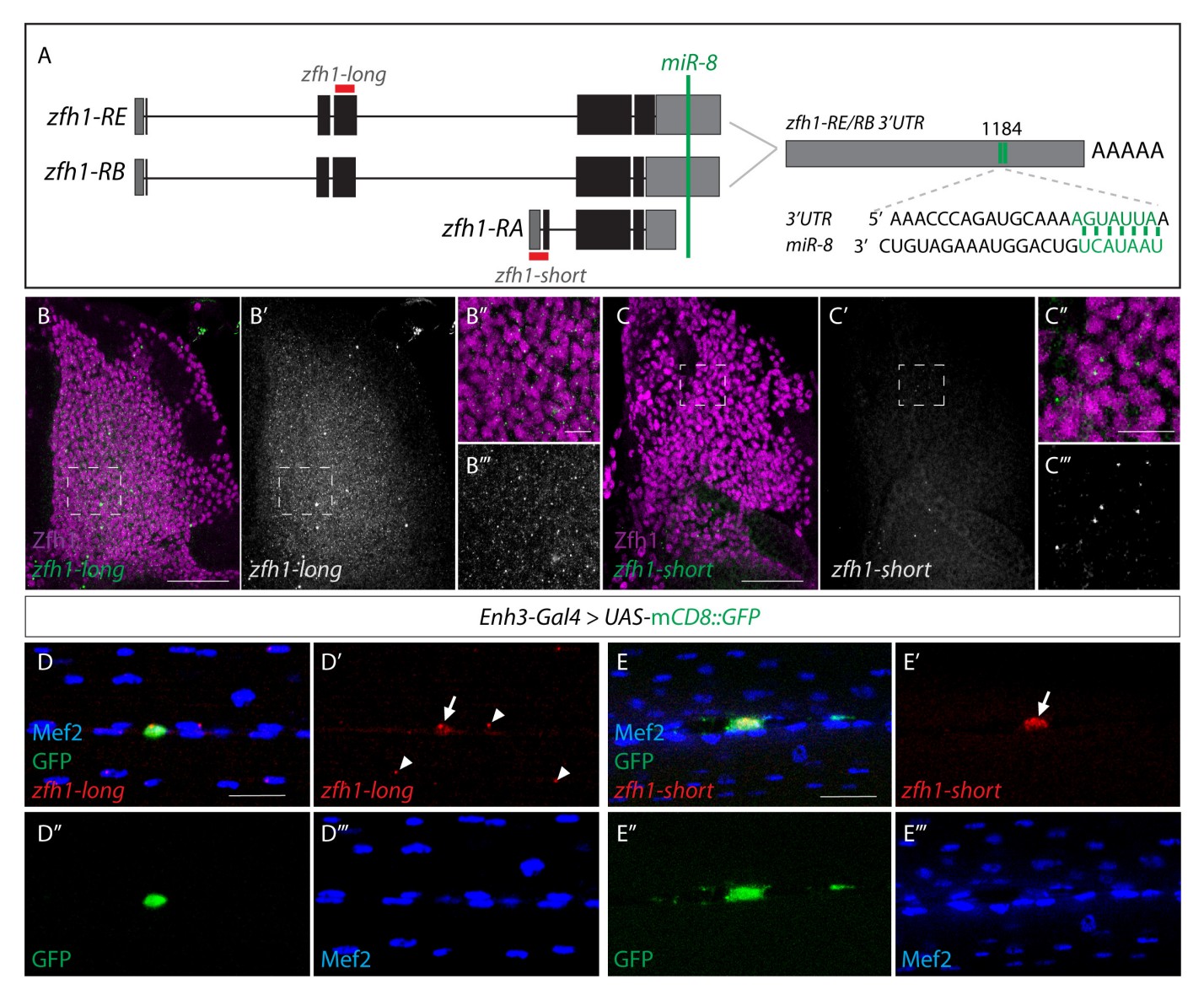

**Figure 7.** Transcriptional dynamics of *zfh1* isoforms in MPs and pMPs. (**A**) Schematic representation of *zfh1* isoforms. *zfh1-short* (*zfh1-RA*) is initiated from a different transcription start site and has shorter 3'UTR that lacks the target site for *miR-8* (Green; *Antonello et al., 2015*) present in *zfh1-long* isoforms (*zfh1-RB, zfh1-RE*); the position of the *miR-8* seed sites in *zfh1-long* 3' UTR are depicted. Non-coding exons and coding exons are depicted by grey and black boxes respectively, red lines indicate the probes used for FISH experiments in B-E. (**B-C**) *zfh1-long* is present uniformly in MPs. (n > 10 wing discs from two replicates; B, Green and B', White) whereas *zfh1-short* is only detected in a few MPs (n > 15 wing discs from three replicates; C, Green and C', White), detected by *in situ hybridisation* in wild type third instar wing discs stained for Zfh1 (Purple). Scale bars: 10 μM. (**B' B''**, **C' C''**) Higher magnifications of boxed regions (Scale bars: 50 μM). (**D-E**) In adult IFMs *zfh1-long* is detected in the pMPs (n = 11 pMPs from two replicates, arrow in D') and in some differentiated nuclei located in their vicinity (arrowheads in D') whereas *zfh1-short* is only present in pMPs (n = 15 pMPs from two replicates, arrow in E'). *Enh3* expression (Green, *Enh3-Gal4 > UAS-mCD8GFP*) labels adult pMPs and Mef2 (Blue) labels all muscle nuclei. Scale bars: 20 μM.

DOI: https://doi.org/10.7554/eLife.35954.012

The following figure supplement is available for figure 7:

**Figure supplement 1.** Zfh1-short isoform is capable of blocking muscle differentiation.
DOI: https://doi.org/10.7554/eLife.35954.013

lower levels of *zfh1-long* (*Figure 7D–E'''*) were present in the pMPs. Indeed, *zfh1-short* was only present in the pMPs whereas dots of *zfh1-long* hybridization were also detected in some differentiated nuclei (with high level of Mef2) (*Figure 7*). Thus, *zfh1-short* is expressed in a few larval MPs and is then detected at highest levels specifically in the adult pMPs but is not transcribed in adult muscle nuclei. Since *zfh1-short* is not susceptible to regulation by *miR-8*, its specific transcription may therefore be determinant for maintaining high levels of Zfh1 in a subset of progenitors and enable them to escape differentiation.

The model predicts that Zfh1-short will counteract the myogenic program in a similar manner as previously described for Zfh1-long, which antagonizes Mef2 function (*Siles et al., 2013*). Premature expression of Mef2 (using *1151* Gal4) induces precocious differentiation of larval MPs, evident by ectopic expression of *MHC-LacZ* and a reduction of MPs (*Figure 7—figure supplement 1* and ref). Expression of Zfh1-short was able to counteract the effects of Mef2, suppressing the precocious muscle differentiation phenotype and restoring the normal morphology of MPs. Thus, Zfh1-short retains the capacity to block Mef2 induced muscle differentiation (*Figure 7—figure supplement 1*).

### *zfh1-short* isoform transcription requires Notch activity in adult pMPs

Expression of *zfh1-short* (*zfh1-RA*) is specifically retained in adult MPs (*Figure 7E*) where it may be critical for maintaining their progenitor status. Mechanisms that ensure this isoform is appropriately transcribed could involve Notch signaling, which is necessary for normal levels of Zfh1 expression in MPs (*Figure 4*). If this is the case, expression of a constitutively active Notch (NotchΔECD) should up-regulate *zfh1-short* transcripts in the MPs at larval stages when their expression is normally low. In agreement, expression of active Notch in the progenitors (*1151-Gal4 > NotchΔECD*) significantly increased the proportions of cells transcribing *zfh1-short* (*Figure 8A–B'* and *C*).

To address whether Notch is necessary for *zfh1-short* transcription in adult pMPs, we specifically depleted Notch levels after eclosion (using *Enh3-Gal4* in combination with *tubGal80^{ts}* to drive *Notch RNAi*; *Figure 8E–F*). Consistent with expression of *zfh1-short* being dependent on Notch activity, the levels of *zfh1-short* were significantly reduced in adult pMPs when Notch was down-regulated in this way (*Figure 8E–F and D*). Conversely, expression of *zfh1-long* isoform was less affected (*Figure 8—figure supplement 1*), suggesting other inputs besides Notch help to sustain *zfh1-long* in pMPs. Nevertheless, Mef2 accumulated to higher levels than normal in the adult pMPs, under these *Notch RNAi* conditions, indicating that Notch activity helps prevent their differentiation (*Figure 8E', F' and H*), most likely through sustaining a higher level of Zfh1 expression via its regulation of *zfh1-short*.

The increased level of Mef2 in the pMPs following Notch-depletion suggests they are losing their progenitor status and becoming differentiated. This forced differentiation of the pMPs would deplete the progenitor population and so should compromise muscle maintenance and repair. In agreement there was a significant reduction of the number of nuclei per muscle after ten days of Notch down-regulation in the pMPs (*Figure 8I*). These results are reminiscent of the targeted *zfh1* down-regulation in the adult pMPs (*Figure 3G–H and I*), where high level of Mef2 was also prematurely detected (*Figure 8G–G' and H*). Taken together, the data indicate that persistent Notch activity is required to maintain *zfh1-short* expression in pMPs, which protects the pMPs from differentiating by ensuring that sufficient Zfh1 protein is present to prevent Mef2 accumulating.

## Discussion

A key property of adult stem cells is their ability to remain in a quiescent state for a prolonged period of time (*Li and Clevers, 2010*). Investigating the maintenance of Drosophila MPs we have uncovered an important new regulatory logic, in which a switch in RNA isoforms enables a sub-population of cells to escape *miRNA* regulation and so avoid the differentiation program. At the same time, analyzing expression of a pivotal player in this regulatory loop, *zfh1*, has revealed how this mechanism sustains a population of persistent progenitors associated with adult muscles in *Drosophila*, that appear analogous to mammalian satellite cells (*Figure 9*).

### Zfh1 maintains a population of satellite-like cells in adult drosophila

Until recently, the fly has been thought to lack a persistent muscle stem cell population, leading to speculation about how its muscles could withstand the wear and tear of its active lifestyle. Now it

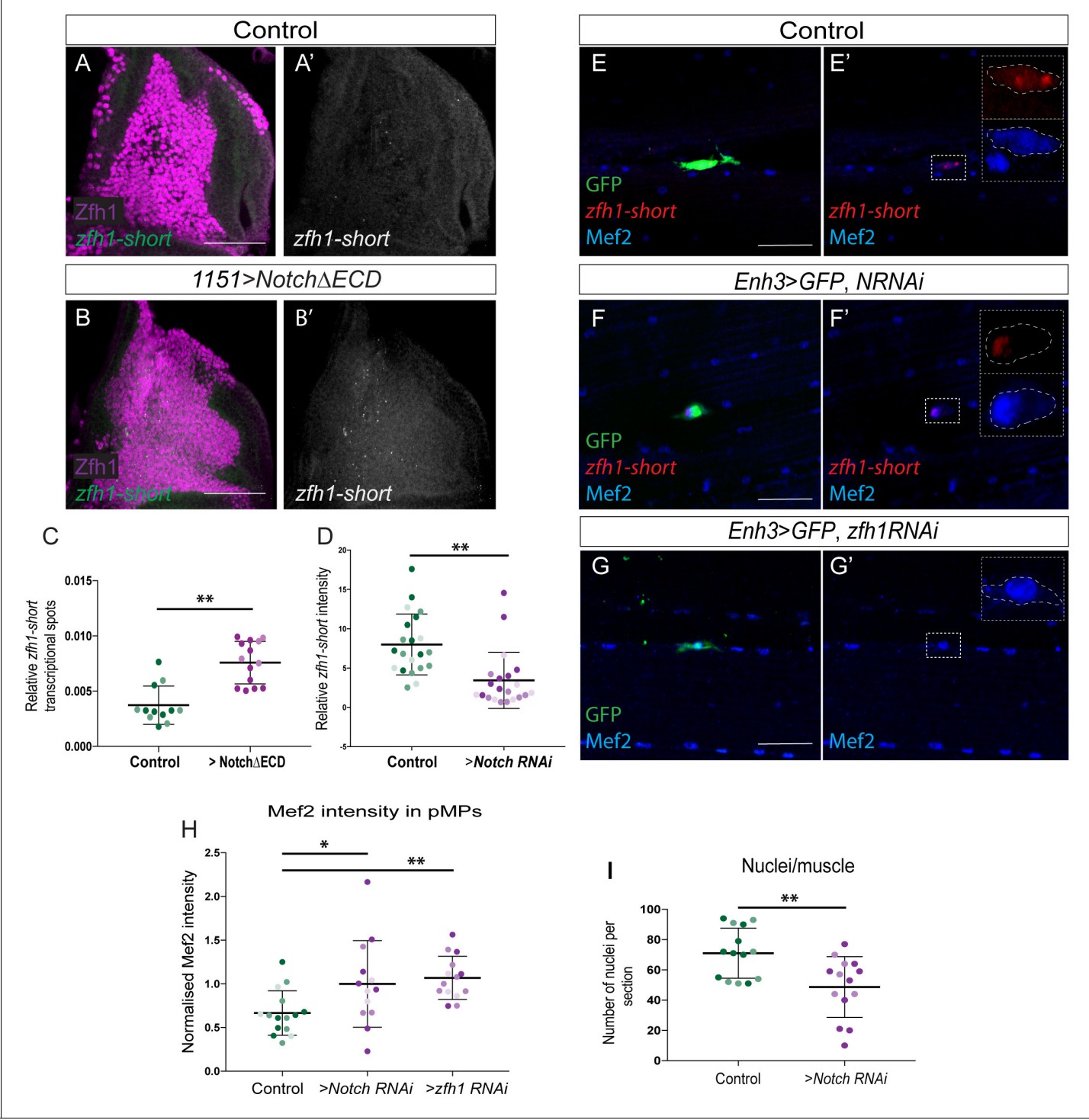

**Figure 8.** *zfh1-short* regulation by Notch is important to maintain muscle homeostasis. (**A-C**) Expression of an activated Notch (*1151-Gal4 > UAS-NΔECD*) in MPs induces ectopic *zfh1-short* transcription. In situ hybridisation detecting *zfh1-short* (Green) in MPs (Zfh1, Purple) with wild type (**A-A'**) or elevated Notch activity (**B-B'**). Scale bars: 50 μM. (**C**) Quantification showing significant increase in *zfh1-short* transcriptional dots upon Notch up regulation, relative to total number of MPs (**p<0.01, Student t-test; n = 14 wing discs for each genotype, light and dark shading indicates data points from two independent replicates). (**D-H**) Notch depletion leads to a severe decrease in *zfh1-short* (Red) (**E'-F' and D**) in pMPs (Green; *Enh3-Gal4; UAS-mCD8GFP > UAS Notch-RNAi; tubGal80ts*) and to an increase in Mef2 levels (Blue) (**E-F' and H**). (**G-G' and H**) *zfh1* depletion in the pMPs (*Enh3-Gal4; UAS-mCD8GFP > UASzfh1 RNAi; tubGal80ts*) leads to an increase in Mef2 levels (Blue). Scale bars: 25 μM. Quantifications of *zfh1-short* (**D**) and Mef2 (**H**) in the indicated conditions show that the levels are significantly different (D, n = 21 pMPs for each genotype (**p<0.01); H, n = 15 pMPs for Control

*Figure 8 continued on next page*

*Figure 8 continued*

RNAi and n = 13 pMPs for *Notch RNAi* (*p<0.05) and n = 14 pMPs for *zfh1 RNAi* (**p<0.01)). In each condition light, dark and intermediate shading indicates data points from three independent replicates. (I). Prolonged *Notch* depletion in the pMPs (*Enh3-Gal4; Gal80ts > UAS Notch RNAi*) affects the muscle homeostasis. (**p<0.01, n = 14 for each genotype, light and dark shading indicates data points from two independent replicates).

DOI: https://doi.org/10.7554/eLife.35954.014

The following figure supplement is available for figure 8:

**Figure supplement 1.** Notch activity is not necessary for *zfh1-long* (Red) transcription (**A'-B' and C**) in pMPs (Green; *Enh3-Gal4; UAS-mCD8GFP > white* RNAi).

DOI: https://doi.org/10.7554/eLife.35954.015

emerges that expression of Zfh-1 identifies a population of muscle-associated cells in the adult that retain progenitor-like properties (*Figure 9* and *Chaturvedi et al., 2017*). Indeed we have found that Zfh-1 is critical to prevent these progenitor cells from differentiating. Its expression in the persistent adult 'satellite-like' cells is dependent on a specific Zfh1 enhancer, which is directly regulated by Notch. Activity of Notch is important for maintaining Zfh1 expression and hence is required to sustain the progenitor status of these cells, similar to the situation in mammalian satellite cells, which require Notch activity for their maintenance (*Mourikis and Tajbakhsh, 2014*; *Bjornson et al., 2012*).

Using lineage-tracing method we showed that adult Zfh1 +ve cells, in normal conditions, provide new myoblasts to the fibers. Furthermore, conditional down regulation of *zfh1* led adult pMPs to

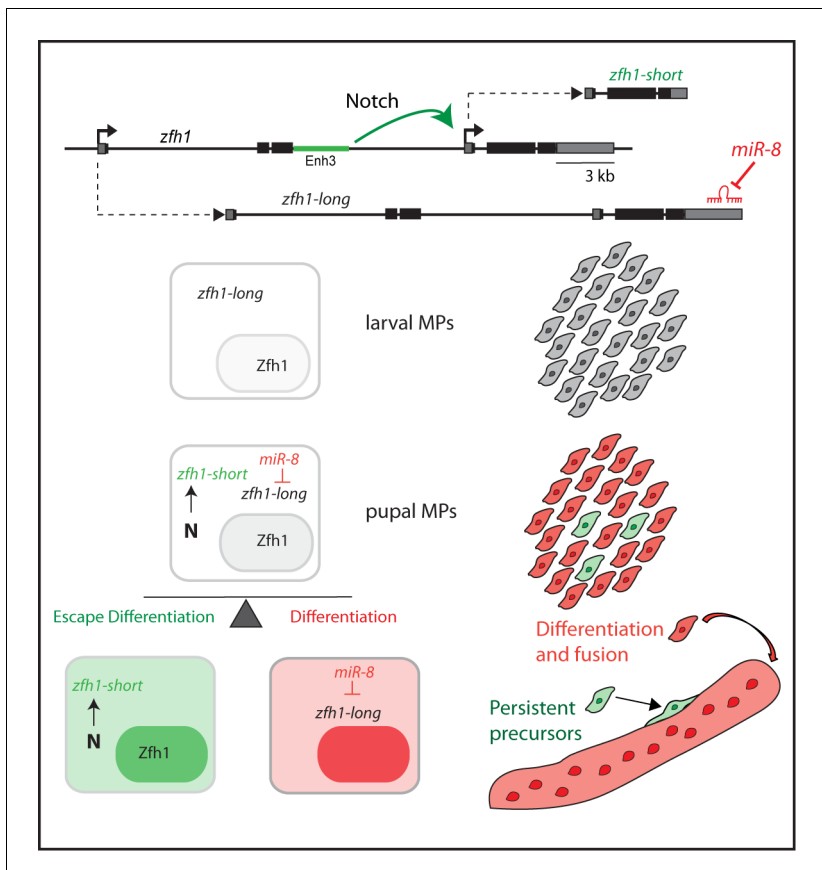

**Figure 9.** Model summarizing the role of alternate *zfh1* isoforms in the maintenance of adult pMPs. *zfh1-long* (Grey) is expressed in all MPs at larval stage. Silencing of *zfh1-long* by *miR-8* (Red) facilitates the MPs differentiation. *zfh1-short* (Green) transcription is driven and maintained in pMPs by a Notch responsive element (*Enh3*, Green rectangle), which may also contribute to *zfh1-long* regulation. Because *zfh1-short* is insensitive to *miR-8*, Zfh1 protein is maintained in pMPs, enabling them to escape differentiation and persist as MPs in the adult.

DOI: https://doi.org/10.7554/eLife.35954.016

enter differentiation and resulted in flight defects, evident by a held out wing posture. These results demonstrate that Zfh-1 is necessary to maintain these progenitors and that, similar to vertebrate satellite cells, the Zfh1 +ve progenitor cells contribute to the adult muscles homeostasis. Others have recently shown that the pMPs are expanded in conditions of muscle injury where they are likely to contribute to repair (*Chaturvedi et al., 2017*). Thus the retention of a pool of progenitor cells may be critical to maintain the physiological function of all muscles in all organism types, as also highlighted by their identification in another arthropod *Parahyle* (*Alwes et al., 2016*; *Konstantinides and Averof, 2014*). *Drosophila* notably differs because their satellite-like cells do not express the Pax3/Pax7 homologue (*gooseberry*; data not shown), considered a canonical marker in mammals and some other organisms (*Chang and Rudnicki, 2014*). Nor do they express the pro-myogenic bHLH protein Twist (data not shown), which is present in the muscle progenitors in the embryo (*Bate et al., 1991*). Instead, Zfh1 appears to fulfill an analogous function and it will be interesting to discover how widespread this alternate Zfh1 pathway is for precursor maintenance. Notably, the loss of ZEB1 in mice accelerates the temporal expression of muscle differentiation genes (e.g. MHC) suggesting that there is indeed an evolutionary conserved function of Zfh1/ZEB in regulating the muscle differentiation process (*Siles et al., 2013*). This lends further credence to the model that Zfh1 could have a fundamental role in preventing differentiation that may be harnessed in multiple contexts.

## Switching 3' UTR to protect progenitors from differentiation

Another key feature of *zfh-1* regulation that is conserved between mammals and flies is its sensitivity to the miR-200/miR-8 family of miRNAs (*Antonello et al., 2015*; *Brabletz and Brabletz, 2010*). This has major significance in many cancers, where loss of *miR-200* results in elevated levels of ZEB1 promoting the expansion of cancer stem cells, and has led to a widely accepted model in which the downregulation of Zfh1 family is necessary to curb stem-ness (*Brabletz and Brabletz, 2010*). This fits with our observations, as we find that *miR-8* is upregulated during differentiation of the MPs and suppresses Zfh1 protein expression. Critically however, only some RNA isoforms, *zfh1-long*, contain seed sites necessary for *miR-8* regulation (*Antonello et al., 2015*). The alternate, *zfh1-short,* isoform has a truncated 3'UTR that lacks the *miR-8* recognition sequences and will thus be insensitive to *miR-8* regulation. Significantly, this *zfh1-short* isoform is specifically expressed in MPs that persist into adulthood and hence can help protect them from *miR-8* induced differentiation during the pupal phases when both are co-expressed. However, the pMPs remain sensitive to forced *miR-8* expression in the adult, suggesting the levels of Zfh1 are finely tuned by the expression of both *zfh1-long* and *zfh1-short*. This could be important to enable the differentiation of the MP progeny. Furthermore, the fact that Notch activity strongly promotes *zfh1-short* expression could explain how an elevated level of Notch signaling promotes expansion of pMPs following injury, as observed by others (*Chaturvedi et al., 2017*).

Together the data suggest a novel molecular logic to explain the maintenance of *Drosophila* satellite-like cells. This relies on the expression of *zfh1-short,* which, by being insensitive to *miR-8* regulation, can sustain Zfh1 protein production to protect pMPs from differentiation (*Figure 9*). It also implies that Notch preferentially promotes the expression of a specific RNA isoform, most likely through the use of an alternate promoter in *zfh1*. Both of these concepts have widespread implications.

Alternate use of 3'UTRs, to escape miRNA regulation, is potentially an important mechanism to tune developmental decisions. Some tissues have a global tendency to favor certain isoform types, for example, distal polyadenylation sites are preferred in neuronal tissues (*Zhang et al., 2005*). Furthermore, the occurrence of alternate 3'UTR RNA isoforms is widespread (>50% human genes generate alternate 3'UTR isoforms) and many conserved *miR* target sites are contained in such alternate 3'UTRs (*Tian and Manley, 2013*; *Sandberg et al., 2008*). Thus, similar isoform switching may underpin many instances of progenitor regulation and cell fate determination. Indeed an isoform switch appears to underlie variations in Pax3 expression levels between two different populations of muscle satellite cells in mice, where the use of alternative polyadenylation sites resulted in transcripts with shorter 3'UTRs that are resistant to regulation by *miR-206* (*Boutet et al., 2012*). The selection of alternate 3'-UTRs could ensure that protein levels do not fall below a critical level (*Yatsenko et al., 2014*), and in this way prevent differentiation from being triggered.

The switch in *zfh1* RNA isoforms is associated with Notch-dependent maintenance of the persistent adult MPs. Notably, *zfh1-short* is generated from an alternate promoter, as well as having a truncated 3'UTR, which may be one factor underlying this switch. Studies in yeasts demonstrate that looping occurs between promoters and polyadenylation sites, and that specific factors recruited at promoters can influence poly-A site selection (*Lamas-Maceiras et al., 2016*; *Tian and Manley, 2013*). The levels and speed of transcription also appear to influence polyA site selection (*Proudfoot, 2016*; *Tian and Manley, 2013*; *Pinto et al., 2011*). If Notch mediated activation via *Enh3* favors initiation at the *zfh1-short* promoter, this could in turn influence the selection of the proximal adenylation site to generate the truncated *miR-8* insensitive UTR. The concept that signaling can differentially regulate RNA sub-types has so far been little explored but our results suggest that is potentially of considerable significance. In future it will be important to investigate the extent that this mechanism is deployed in other contexts where signaling coordinates cell fate choices and stem cell maintenance.

## Materials and methods

### Key resources table

| Reagent type (species) or resource | Designation | Source or reference | Identifiers | Additional information |
|---|---|---|---|---|
| Gene (D. melanogaster) | zfh1 | NA | FLYB:FBgn0004606 | |
| Gene (D. melanogaster) | Notch | NA | FLYB:FBgn0004647 | |
| Gene (D. melanogaster) | miR-8 | NA | FLYB:FBgn0262432 | |
| Genetic reagent (D. melanogaster) | Enh3-Gal4 | Janelia Research Campus | BDSC: 49924, FLYB: FBtp0059625 | FlyBase symbol: P{GMR35H09-GAL4}attP2 |
| Genetic reagent (D. melanogaster) | zfh1 RNAi (kk 103205) | Vienna Drosophila RNAi Center | VDRC: 103205 | |
| Genetic reagent (D. melanogaster) | miR-8-Gal4 | Kyoto Stock Center | DGRC: 104917 | Genotype: y[*] w[*]; P{w[+mW.hs]=GawB}NP5247/CyO, P{w[-]=UAS lacZ.UW14}UW14 |
| Genetic reagent (D. melanogaster) | UAS-miR-8-Sp | Bloomington Stock Center | BDSC: 61374, FLYB: FBst0061374 | Genotype: P{UAS-mCherry.mir-8.sponge.V2}attP40/CyO; P{UAS-mCherry.mir-8.sponge.V2}attP2 |
| Genetic reagent (D. melanogaster) | Enh3-GFP | This paper | | |
| Genetic reagent (D. melanogaster) | ΔEnh3 | This paper | | |
| Genetic reagent (D. melanogaster) | UAS-zfh1-short | This paper | | UAS-Zfh1-short construct provided by BDGP, Clone # UF5607 |
| Genetic reagent (D. melanogaster) | Notch[NRE]-GFP | Sarah Bray (Cambridge, UK) | | |
| Antibody | anti-Zfh1 | Ruth Lehmann (New York, USA) | | |
| Antibody | anti-Mef2 | Eileen Furlong (Heidelberg, Germany) | | |
| Recombinant DNA reagent | pCFD4 | Addgene | Addgene # 49411 | |
| Recombinant DNA reagent | pDsRed-attP | Addgene | Addgene # 51019 | |

### Drosophila genetics

All *Drosophila melanogaster* stocks were grown on standard medium at 25°C. The following stains were used: *w^118* as wild type (wt), *UAS-white-RNAi* as control for *RNAi* experiments (BL35573), *UAS-zfh1-RNAi* (VDRC: KK103205, TRiP: BL29347), *zfh1* deficiency (BL7917), *Mef2-Gal4*

(*Ranganayakulu et al., 1996*), UAS-Mef2 (*Cripps et al., 2004*), UAS-G-Trace (BL28281), UAS-Notch-RNAi (BL7078), Notch[NRE]-GFP (*Simón et al., 2014*), UAS-NotchΔECD (*Chanet et al., 2009*; *Fortini and Artavanis-Tsakonas, 1993*; *Rebay et al., 1993*), miR-8-Gal4 (*Karres et al., 2007*), UAS-miR-8 (*Vallejo et al., 2011*), UAS-miR-8-Sp (BL61374 and *Fulga et al., 2015*), UAS-Scramble-SP (BL61501), UAS-mCD8::GFP (BL5137), UAS-GFPnls (BL65402), UAS-Src::GFP (*Kaltschmidt et al., 2000*), MHC-lacZ (*Hess et al., 2007*), 1151-Gal4 (*Anant et al., 1998*), miR-8-sensor-EGFP (*Kennell et al., 2012*), CG9650-LacZ (*Ahmad et al., 2014*), UAS-Reaper (BL5824). Enhancer-Gal4 lines described in *Figure 2* and *Figure 2—figure supplement 1* are either from Janelia FlyLight (http://flweb.janelia.org) or Vienna Tiles Library (http://stockcenter.vdrc.at/control/main).

RNAi experiments were conducted at 29°C. For adult specific manipulations in pMPs *tubGal80ts* (*McGuire et al., 2003*) was used to limit *Enh3-Gal4* expression to a defined period of time. Crosses were kept at 18°C and eclosed adults were shifted to 29°C until dissection.

## Immunohistochemistry and in situ hybridization

Immunofluorescence stainings of wing discs were performed using standard techniques. Dissection and staining of the pupal muscles was performed according to (*Weitkunat and Schnorrer, 2014*). Adult muscles were prepared and stained as described in (*Hunt and Demontis, 2013*). The following primary antibody were used: Rabbit anti-Zfh1 (1:5000, a gift from Ruth Lehmann, New York, USA), Mouse anti-Cut (1:20, DSHB), Rabbit anti-β3-Tubulin (1:5000, a gift from Renate Renkawitz-Pohl, Marburg, Germany), Rat anti-Tropomyosin (1:1000, Abcam, ab50567), Goat anti-GFP (1:200, Abcam, ab6673), Rabbit anti-Ds-Red (1:25; Clontech, 6324496), Rabbit anti-Mef2 (1:200, a gift from Eileen Furlong, Heidelberg, Germany), Mouse anti-P1 (1:20, a gift from István Andó, Szeged, Hugary), Mouse anti-pH3 (1:100, Cell Signaling Technology, #9706), Mouse anti-β-Gal (1:1000, Promega, Z378A), Alexa-conjugated Phalloidin (1:200, Thermo fisher, Waltham/Massachusetts), Rat anti-Dcad2 (1:200, DSHB). In situ experiments were carried out according to Stellaris-protocols (https://www.biosearchtech.com/assets/bti_custom_stellaris_drosophila_protocol.pdf). Antibodies were included to the overnight hybridization step (together with the probes). *zfh1* probes were generated by Biosearch Technologies. The sequence used for *zfh1-short* probe span 393 bp of the first *zfh1-RA* exon, for *zfh1-long* probe, the sequence of the third exon (711 bp) common to both *zfh1-RB* and *zfh1-RE* was used (see *Figure 5*).

## Construction of transgenic lines and mutagenesis

For Enh3-GFP reporter line, the genomic region chr3R: 30774595..30778415 (Enh3/GMR35H09) according to Flybase genome release r6.03 was amplified using *yw* genomic DNA as template. *Enh3* fragment was then cloned into the pGreenRabbit vector (*Housden et al., 2012*). For Enh3[mut]-GFP line, two Su(H) biding sites were predicted within *Enh3* sequence using Patser (*Hertz and Stormo, 1999*) and mutated by PCR based approach with primers overlapping the Su(H) sites to be mutated with the following sequence modifications: Su(H)1 AG<u>TGGGA</u>A to AG<u>GTGTGA</u> and Su(H) 2 T<u>TCTCACA</u> to T<u>GTTTG</u>CA. Both constructs were inserted into an AttP located at 68A4 on chromosome III by injection into *nos-phiC31-NLS; attP2* embryos (*Bischof et al., 2007*). The UAS-zfh1-short transgenic line (*Figure 7—figure supplement 1*) was similarly generated using phiC mediated integration of an AttB plasmid carrying *UAS-zfh1-short* (BDGP Clone # UF5607) into an AttP site at position 25C7. The transgene produced detectable nuclear Zfh1 protein when crossed to Gal4 driver lines (data not shown). All constructs were fully sequenced and analyzed prior to injection.

## CRISPR/Cas9 genome editing

CRISPR mediated deletion of *Enh3* was performed according to (*Port et al., 2014*). For generating guide RNAs, two protospacers were selected (sgRNA1 GCATTCCGCAGGTTTAGTCAC and sgRNA2 GCGATAACCCGGCGACCTCC) flanking 5' and 3' *Enhancer-3* regions, (http://www.flyrnai.org/crispr/). The protospacers were cloned into the tandem guide RNA expression vector pCFD4 (Addgene #49411) (http://www.crisprflydesign.org/wp-content/uploads/2014/06/Cloning-with-pCFD4.pdf). For the homology directed repair step, two homology arms were amplified using *yw* genomic DNA as template with the following primers (Homology arm1: Fwd. 5' GCGCGAATTCGGGCTAAACGCCAGATAAGCG 3' Rev. 5' TTCCGCGGCCGCCACTGGGATTCCACGGCTTTTCG 3'– Homology arm 2: Fwd. 5' GGTAGCTCTTCTTATATAACCCGGCGACCTCCTCG3'- Rev. 5'GGTAGC

TCTTCTGACC GGACGAAAAACTAGCGACC) and cloned into the pDsRed-attP (Addgene #51019) vector (http://flycrispr.molbio.wisc.edu/protocols/pHD-DsRed-attP). Both constructs were injected into *nos-Cas9* (BL54591) embryos. Flies with ΔEnh3 were identified via the expression of the Ds-Red in the eyes and confirmed with sequencing of PCR fragment spanning the deletion. Δ*Enh3* flies were then crossed to strains carrying a deletion (BL7917), which removes *zfh1*. None of the tranheterozygote animals survived to adults confirming that Δ*Enh3* lethality maps to the *zfh1* locus.

## Microscopy and data analysis

Samples were imaged on Leica SP2 or TCS SP8 microscopes (CAIC, University of Cambridge) at 20X or 40X magnification and 1024/1024 pixel resolution. Images were processed with Image J and assembled with Adobe Illustrator. Quantification of fluorescence signal intensities was performed with Image J software. In each case the n refers to the number of individual specimens analyzed, which were from two or more independent experiments. For experiments to compare and measure expression levels, samples were prepared and analyzed in parallel, with identical conditions and the same laser parameters used for image acquisition. For each confocal stack a Sum slices projections was generated. Signal intensities were obtained by manually outlining the regions of interest, based on expression of markers, and measuring the average within each region. The values were then normalized to similar background measurements for each sample. In *Figure 8* the number of transcriptional *zfh1-short* dots was counted manually with Image J and normalized to the total number of nuclei (Zfh1 staining), which was determined by a Matlab homemade script. Graphs and statistical analysis were performed with Prism seven using unpaired t-test. Error bars indicate standard error of the mean. Further statistical details of experiments can be found in the figure legends.

## Quantitative RT PCR

30 Wing Imaginal discs from each genotype were dissected and RNA isolated using TRIzol (Life technologies). Quantitative PCR were performed as described (*Krejcí and Bray, 2007*). Values were normalized to the level of *Rpl32*. The following primers were used. *Rpl32*, Fwd 5'-ATGCTAAGCTG TCGCACAAATG-3' and Rev 5'-GTTCGATCCGTAACCGATGT-3'. *zfh1* Fwd 5'- GTTCAAGCACCACC TCAAGGAG-3' and Rev 5'- CTTCTTGGAGGTCATGTGGGAGG-3'. (Product common to all three *zfh1* isoforms).

## Acknowledgements

We acknowledge the Bloomington Stock Center, the VDRC Stock Center and the Developmental Studies Hybridoma Bank for *Drosophila* strains and antibodies. We thank Ruth Lehmann, Renate Renkawitz-Pohl, Eileen Furlong and István Andó for antibodies. We thank Eva Zacharioudaki, Alain Vincent and Michalis Averof for critical reading of the manuscript and other members of SJB lab for valuable discussion; and we are grateful to Hannah Green and Juanjo Perez-Moreno for advice on muscle preparations. This work was funded by a program grant from the MRC to SJB and by an EMBO Long Term Fellowship for HB.

## Additional information

### Funding

| Funder | Grant reference number | Author |
| --- | --- | --- |
| Medical Research Council | MRL007177/1 | Sarah Bray |
| European Molecular Biology Organization | ALTF-325-2013 | Hadi Boukhatmi |

The funders had no role in study design, data collection and interpretation, or the decision to submit the work for publication.

## Author contributions
Hadi Boukhatmi, Sarah Bray, Conceptualization, Supervision, Funding acquisition, Validation, Investigation, Visualization, Methodology, Writing—original draft, Project administration, Writing—review and editing

## Author ORCIDs
Sarah Bray http://orcid.org/0000-0002-1642-599X

## Decision letter and Author response
Decision letter https://doi.org/10.7554/eLife.35954.019
Author response https://doi.org/10.7554/eLife.35954.020

## Additional files
### Supplementary files
• Transparent reporting form
DOI: https://doi.org/10.7554/eLife.35954.017

### Data availability
All data generated or analysed during this study are included in the manuscript and supporting files.

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
