## [Decision Letter]

[Editors’ note: a previous version of this study was rejected after peer review, but the authors submitted for reconsideration. The first decision letter after peer review is shown below.]

Thank you for submitting your work entitled "A population of adult satellite-like cells in *Drosophila* is maintained through a switch in RNA-isoforms" for consideration by *eLife*. Your article has been favorably evaluated by a Senior Editor and three reviewers, one of whom is a member of our Board of Reviewing Editors. The reviewers have opted to remain anonymous.

Our decision has been reached after consultation between the reviewers. Based on these discussions and the individual reviews below, we regret to inform you that your work will not be considered further for publication in *eLife*.

The reviewers agree that this is a very interesting study proposing a conceptually novel regulatory logic in muscle stem cell development. However, all three reviewers conclude that some of the key pieces of evidence are still too preliminary and do not yet firmly support the conclusions made. As you will see from the attached reviews, the main weaknesses include a) the absence of clonal evidence showing that the satellite-like cells are contributing to muscle homeostasis, b) uncertainties regarding the role of Zfh1 short in these cells and their precursors, c) the lack of data demonstrating the developmental expression dynamics of *miR-8*, Enh3-GFP, and Zfh1 within the satellite-like cells and their precursors, and d) the insufficient characterization of *mir-8* function within these cells that would support the biological relevance of this regulatory logic. As the reviewers expect that the work to address these issues would take significantly longer than the two months requested for a resubmission, we are returning the manuscript to you and hope that the reviews will provide helpful guidance for further strengthening the study.

*Reviewer #1:*

Boukhatmi and Bray present intriguing data suggesting the existence of muscle stem cells in adult *Drosophila* that may function like the satellite cells in vertebrates during muscle repair upon injury and aging. The authors propose a major role of the transcription factor Zfh1 in keeping these cells undifferentiated. They extend the regulatory pathway further by findings indicating that a short isoform of zfh1 is being induced by Notch/Su(H) in these satellite-like cells, which in contrast to the long form is not targeted by the microRNA miR-8 that could otherwise prevent expression of Zfh1 in them. Through this mechanism, Notch could stabilize the undifferentiated state of these satellite-like cells.

In principle, these are very important and novel findings that would have broad relevance in the muscle stem cell field. However, as discussed below, there are still some major gaps and inconsistencies in the data and models, which means that the evidence for functional muscle satellite cells in *Drosophila* in this study is largely circumstantial but without solid proof, and some aspects of the proposed regulatory pathway are questionable. However, if the authors were able to fill in these gaps convincingly the manuscript would be highly appropriate for publication in *eLife*.

1) Currently, the authors do not have any direct evidence that the observed cells act as muscle stem cells that can fuse with muscles for repairing damage. The Enh3-Gal4>G-TRACE lineage data with GFP (Figure 2) only show that both the muscles and the satellite-like cells are derived from adepithelial cells of the wing discs, as would be expected. The demonstration of proliferation is interesting, but not proof for their potential to achieve muscle repair. The crucial experiment would be to perform this experiment conditionally, such that GFP becomes only expressed in the satellite-like cells and their descendants after hatching, and then look for GFP-positive nuclei *within* muscles upon aging or injury. I realize there may be some diffusion of GFP in the syncytia but nevertheless, the bulk of GFP should be imported into the nuclei that express GFP RNA.

Related to this point, in Figure 7 the authors show that, upon knockdown of *zfh1* after hatching and 8 days of further aging of the flies, their IFMs contain ca. 20% fewer nuclei, which they take to mean that normally that many satellite cells fuse to repair the aging muscles during this time period. This would be a surprisingly large percentage and an important result. If true, the conditional lineage tracing experiment requested above should show ca. 20% GFP-positive nuclei in IFM cross sections of aged flies similar to those shown in Figure 7. Alternatively, could the reduction in muscle nuclei be caused by leaky earlier dsRNA expression during muscle development?

2) Given the recurring off-target and other artifacts with RNAi, the differences in phenotypes with different *zfh1* RNAi lines are a bit worrisome. Such effects could also explain the different phenotypes with the DeltaEnh3 allele and with forced miR-8 expression. In particular, whereas with the KK line there is some premature differentiation while the undifferentiated bulk of adepithelial cells look normal, with the GD line all of the adepithelial cells look morphologically abnormal but none are differentiated. First, the authors should verify by antibody stainings that Zfh1 levels are indeed strongly reduced in these experiments. Second, they should rule out that the effect with the KK line is caused the known second site insertion that activates the Hippo pathway, which is present in a significant portion of the KK lines (Vissers et al., Nature Comm. 2016). Third, they should use the Valium10 and -20 lines from TRiP to check for consistent phenotypes. In this context, it is not always clear which one of the lines was used in a particular experiment (e.g., in Figure 7. And does the other line give the same phenotype?).

3) –In the subsection “An alternate short *zfh1* isoform is transcribed in adult MPs” and elsewhere the authors argue repeatedly that Zfh1 expression and thus stem cell maintenance must "evade" regulation by miR-8, which they explain by the expression of the short isoform that lacks miR-8 target sequences. However, in Figure 4 they show that *mir-8* is not even expressed in the satellite-like cells, which means there is actually no need for evasion in these cells. Currently, there is only evidence that *miR-8* is expressed in fully differentiated muscles, which could be downstream of differentiation regulators such as *Mef2* (in line with data from Figure 6—figure supplement 1). So the proposed evasion mechanism is highly speculative and cannot be presented as a fact without any supporting data. Such data could for instance come from G-TRACE experiments with *miR-8-GAL4* that show that *miR-8* is indeed transiently expressed in the precursors of the satellite-like cells.

4) –Subsection “*zfh1* isoform transcription requires Notch activity in adult SCs”, second paragraph, Figure 6, Figure 8—figure supplement 1: The authors ascribe the high *Mef2* levels (i.e., possible entry into differentiation) in satellite cells with depleted Notch to reduced *zfh1*-short. However, as they show, *zfh1*-long is unaffected by Notch depletion and because *mir-8* is absent in these cells (see above), *zfh1* could still suppress differentiation. This would argue for *zfh1*-independent effects of Notch.

*Reviewer #2:*

Bokhatmi and Bray report the presence of a population of satellite-like stem cells in the adult of *Drosophila* that are maintained through a switch of *zfh1* RNA -isoforms that enable escape from microRNA *miR-8/miR-200* control. They authors show that differentiation of MP is associated with expression of *miR-8*, which targets *zfh1*-long RNA thus down-regulating *zfh1* protein. They also map an enhancer (Enh3) region that drives expression of *zfh1* gene in some MPs in the larval wing imaginal discs and the adult MPs. Interestingly, this enhancer is regulated by Su (H) and these authors show very clearly that Notch-Su(H) activates an alternate *zfh1* short isoform in adult MPs, where Notch is also normally activated. *zfh1* short RNA lacks the *miR-8* seed sequence and thus cannot be silenced by *miR-8*. Altogether the data suggest that Notch-driven *zfh1* short allows the *zfh1* to escape down-regulation of *miR-8* and thus terminal differentiation in a subset of larval muscle progenitor cells. The authors also postulate that expression of the alternate *zfh1*-short isoform is a critical part of the regulatory switch to maintain a pool of progenitor "satellite-like" cells in the adult.

I have some critics and specific comments that need clarification:

1) Role of Zfh1 short RNA isoform in the maintenance of muscle progenitors. Some additional experiments, such as gain-of-expression of *zfh1* isoforms, specifically the short form to substantiate the model.

2) Loss of *mir-8* assays is paramount to the model. The *mir-8* sponge should produce phenotypes that substantiate the biological relevance of the regulatory switch to maintain a subset of MPs retaining *zfh1. mir-8* loss would be expected to increase the number of pMPs.

3) Epistatic studies to link *mir-8* overexpression to *zfh1* (gain of *mir-8* along with gain of *zfh1* and/or loss of *mir-8* and loss of *zfh1*). These experiments seem important for substantiating the biological relevance of the regulatory loci discovered.

4) The conclusion that adult Zfh1+ve cells retain capacity to divide and produce progeny should be reinforced with direct data. G-TRACE is not a cell lineage method – it traces the dynamics of a Gal4 expressing cell. It is very likely the authors' conclusion is right but a clonal cell lineage method is more appropriate for this particular issue.

In summary, this is a very interesting study with a conceptually novel regulatory logic that may have broader implications for understanding how other stem cells and cancer stem-like cells may escape terminal differentiation. Nonetheless, some additional data would be required to substantiate that the zfh1 short isoform contributes (or is capable) to the maintenance of satellite-like progenitor cells.

*Reviewer #3:*

In the submitted manuscript H. Boukhatmi and S. Bray analyse an intriguing issue related to the mechanisms of maintenance of persistent Muscle Progenitors (pMPs) in adult *Drosophila* muscles focusing on flight muscles. Considering that identification and functional characterisation of *Drosophila* muscle satellite-like cells that express Zfh-1 has already been reported (open access BioRxiv Chaturvedi et al., manuscript posted 25 Jan 2016 from VijayRaghavan lab) the Boukhatmi and Bray's manuscript carries two new findings: i) pMPs preferentially express a short isoform of *zfh1*, which is devoid of *miR8* seed site and ii) *zfh1* is not only present in pMPs but also plays an important role in their maintenance. Authors also identify and apply a new tool, *zfh1-Enh3-Gal4* that allows them to target/visualize pMPs.

My comments focus on these two aspects, which make this manuscript of potential interest for *eLife* if documented more rigorously.

1) Differential post-transcriptional regulation of *zfh1* isoforms by *miR8* could indeed represent a major regulatory mechanism in setting pMPs population. Authors have tools in their hands to document when this isoform switch takes place. In Figure 2D they show that in 3rd instar larvae Enh3>GFP is largely expressed in Zfh1/Cut positive MPs in the notum region and then they jump to adult stage to show Enh3-GFP restricted to pMPs only. It would be interesting to see when in development this restriction is effective. In other words when the satellite-like pMPs could be identified and visualized for the first time. Time lapse imaging of IFM development in Enh3-GFP pupa (approach developed in Schnorrer lab) could represent an interesting view point as well. The developmental restriction of Enh3-GFP expression should be correlated with miR8 sensor and/or *mir8-Gal4* expression to evaluate whether these two regulatory events are coordinated. In relation to this it would be important to better correlate Enh3 activity with transcriptional regulation of *zfh1*-short isoform. It is unclear whether Enh3 indeed regulates preferentially transcription of *zfh1*-short. Levels of short and long transcripts could be quantified in ΔdeltaEnh3 context in wing discs (instead of Zfh1 protein shown in Figure 2G) and in adult pMPs.

2) Regarding second major finding that *zfh1* is required for maintenance of pMPs it could be important to test functional relevance of the two isoforms. Enh3>miR8 could be applied to deplete pMPs of *zfh1*-long and Enh3>NRNAi to see if effects of N and zfh1RNAi are similar (with experimental setup like in Figure 7). Generation of CRISPRs mutation of *zfh1*-short promoter/TSS site could help in defining function of this isoform.

3) Referring to the role of pMPs in muscle homeostasis one complementary experiment could be to test effects of ablation of these cells in adult flies (Enh3>Rpr).

[Editors’ note: what now follows is the decision letter after the authors submitted for further consideration.]

Thank you for resubmitting your work entitled "A population of adult satellite-like cells in *Drosophila* is maintained through a switch in RNA-isoforms" for further consideration at *eLife*. Your article has been favorably evaluated by Diethard Tautz (Senior Editor) and three reviewers, one of whom is a member of our Board of Reviewing Editors.

The manuscript has been improved but there are some remaining issues that need to be addressed before acceptance, as outlined below:

The main issue concerns the evaluation of phenotypes based on staining intensities. It is still unclear whether each of these is derived from a single batch of staining or from several experimental replicates. In case of the former, this would lower the level of confidence because of inevitable variabilities in the fixation and staining procedures.

So we request either to provide information on the numbers of independent experiments in each case or, if the data came only from single experiments, to perform additional ones to obtain more meaningful statistics.

The comments of the reviewers are attached and you may respond to them as well where you find it necessary. But the main point to be dealt with is the one above.

*Reviewer #1:*

The revised resubmission by Boukhatmi and Bray has addressed the majority of the criticisms of the reviewers. The most important additions include:

1) Experimental evidence by fate mapping and ablation experiments that support the notion that the pMPs serve as satellite-like cells to promote adult muscle homeostasis. The recently published *eLife* paper by Chaturvedi et al. on the same topic is now incorporated and referenced.

2) Immunohistochemical analyses of the time course of *miR-8* vs. Zfh1 expression from pupal to adult stages. Although *miR-8* expression was monitored indirectly via *miR-8-GAL4* driving UAS-nGFP, which due to Gal4 and GFP perdurance likely produces longer-lasting signals than *miR-8* RNA proper, these data do help to strengthen the points made by the authors. The dynamic activity of Enh3 from *zfh1* has been documented in more detail as well.

3) Conditional depletion experiments for Zfh1 in adults, which show a role of zfh1 in maintaining the pMPs and, presumably as a consequence, in adult muscle homeostasis.

4) Conditional depletion experiments for *miR-8* using forced expression of a *miR-8* sponge and, conversely, of *miR-8*, results of which support the mechanistic model proposed by the authors.

Altogether, these data significantly strengthen the story presented in the manuscript.

*Reviewer #2:*

Authors generated impressive set of new data that support and further reinforce their initial findings. In the revised version of the manuscript they also answer all issues raised in my comments.

I would like to congratulate them for this very nicely documented work that provides an example of a how differentiation versus non-differentiation of a cell could be regulated.

The images they provide are convincing but regarding the intensity it would be good to have further information in how they had performed the analysis. Usually, it is a good idea to do flip-out clones so that there is internal control and one can appreciate the different intensity between the mutant cells and the wt surrounding within the same staining. However this might be difficult for some experiment here.

*Reviewer #3:*

This is a much improved manuscript which clarified all my previous concerns and specific questions. The authors make a strong case and provide a mechanistic understanding on how a population of satellite-like stem cells in the adult of *Drosophila* maintain their undifferentiated state through a switch of *zfh1* RNA -isoforms that enable the escape from the *miR-8* microRNA inhibition. The regulatory logic is further sustained in the current revised version and now the authors provide new functional evidence for the requirement of *mir-8* and the *zfh1* short-isoform. Along with the previous results, the new data consolidate the model that differentiation of MP is linked to *miR-8*-mediated inhibition of *zfh1 (zfh1*-long RNA) and that scape from *miR-8 (zfh1*-short) maintains a pool of MPs that sustain muscle homeostasis. The time course analyses of expression of the genes, and the data that *zfh1*-positive cells contribute to differentiated adult muscle are also strong and the images clear and of high quality. In their study, they authors also map the regulation of the *zfh1*-short isoform to Notch, and identified an enhancer (Enh3) region that drives expression of *zfh1* gene in some MPs in the larval wing imaginal discs and the adult MPs. This Enh3-Gal4 has been, and will be a valuable tool for further studies of these cells. The data are consistent with Notch-driving expression of the alternate *zfh1* short isoform at the right time to allow this important stem cell factor to escape down-regulation by *miR-8* in a subset of larval muscle progenitor cells. Some conclusions are derived from indirect data, but the authors explain the technical limitations and provide alternative experimental data that form a consistent model. In summary, the authors provide support for the model that expression of the alternate *zfh1*-short isoform is a critical part of the regulatory switch to select a pool of progenitor "satellite-like" cells that are capable to sustain muscle repair in the adult fly.

---

## [Author Response]

[Editors’ note: the author responses to the first round of peer review follow.]

Reviewer #1:[…] 1) Currently, the authors do not have any direct evidence that the observed cells act as muscle stem cells that can fuse with muscles for repairing damage. The Enh3-Gal4>G-TRACE lineage data with GFP (Figure 2) only show that both the muscles and the satellite-like cells are derived from adepithelial cells of the wing discs, as would be expected. The demonstration of proliferation is interesting, but not proof for their potential to achieve muscle repair. The crucial experiment would be to perform this experiment conditionally, such that GFP becomes only expressed in the satellite-like cells and their descendants after hatching, and then look for GFP-positive nuclei within muscles upon aging or injury. I realize there may be some diffusion of GFP in the syncytia but nevertheless, the bulk of GFP should be imported into the nuclei that express GFP RNA.Related to this point, in Figure 7 the authors show that, upon knockdown of zfh1 after hatching and 8 days of further aging of the flies, their IFMs contain ca. 20% fewer nuclei, which they take to mean that normally that many satellite cells fuse to repair the aging muscles during this time period. This would be a surprisingly large percentage and an important result. If true, the conditional lineage tracing experiment requested above should show ca. 20% GFP-positive nuclei in IFM cross sections of aged flies similar to those shown in Figure 7. Alternatively, could the reduction in muscle nuclei be caused by leaky earlier dsRNA expression during muscle development?

We thank the reviewer for their suggestion and comments. We note however that in our initial experiments we had focused on the perdurance of the RFP signal in the newly born myoblasts to trace their origin, because of the caveats mentioned by the reviewer. However we agree that the question of whether these cells contribute to normal homeostatic muscle repair is an important one, and has not been addressed before. We have taken two strategies to investigate. First as suggested by the reviewer we have used thermosensitive Gal80 (Gal80ts) to conditionally inactivate the Enh3-Gal4 until the adult flies emerge, then raised the temperature to allow Gal4 activity and traced the progeny from this adult-only expression with G-Trace. Specifically we quantified the GFP-positive nuclei present within adult muscles 10 days after they had emerged, and found that around 24% of the nuclei contained GFP (new Figure 3C-D, subsection “Adult Zfh1+ve MP cells contribute to flight muscles”, last paragraph).These GFPpositive nuclei were most concentrated in proximity to the satellite-like cells supporting the model that their progeny contribute to muscle homeostasis. As the reviewer rightly points out, the proportion of cells that we label fits well with the predictions based on the effects from depleting *zfh1* in these cells.

Second, to provide further evidence that the satellite cells are important for normal muscle repair, we performed a genetic ablation by expressing a pro-apoptotic gene, reaper. Strikingly, this led to a similar reduction in muscle nuclei to that from *zfh1* depletion (new Figure 3I, see the aforementioned paragraph). Altogether these data support the conclusion that the pMPs are contributing to muscle maintenance by providing new myoblasts.

2) Given the recurring off-target and other artifacts with RNAi, the differences in phenotypes with different zfh1 RNAi lines are a bit worrisome. Such effects could also explain the different phenotypes with the DeltaEnh3 allele and with forced miR-8 expression. In particular, whereas with the KK line there is some premature differentiation while the undifferentiated bulk of adepithelial cells look normal, with the GD line all of the adepithelial cells look morphologically abnormal but none are differentiated. First, the authors should verify by antibody stainings that Zfh1 levels are indeed strongly reduced in these experiments. Second, they should rule out that the effect with the KK line is caused the known second site insertion that activates the Hippo pathway, which is present in a significant portion of the KK lines (Vissers et al., Nature Comm. 2016). Third, they should use the Valium10 and -20 lines from TRiP to check for consistent phenotypes. In this context, it is not always clear which one of the lines was used in a particular experiment (e.g., in Figure 7. And does the other line give the same phenotype?).

We appreciate the reviewer’s concerns about the specificity of the *zfh1* RNAi KK line that was the primary tool used in our analysis. Following the reviewer’s request we have verified that the KK RNAi line strongly reduces Zfh1 protein levels (new Figure 1—figure supplement 1D, E). We have also taken on board their important suggestion to confirm the results using further independent RNAi lines. We have new data demonstrating that the TRiP Valium10 *zfh1* shRNA gives the same premature differentiation phenotype as *zfh1* RNAi KK (new Figure 1—figure supplement 1F-H). We have clarified in the figure legends which specific RNAi line was used for a given experiment.

With respect to the “differences in phenotypes” we note that there is a correlation between the severity of the phenotype, as measured by ectopic expression of differentiation markers, and the morphology of any residual MP/adepithelial cells. In some discs there is strong Tropomyosin expression in all the cells and there are few with any undifferentiated morphology, in others there are fewer Tropomyosin expressing cells and many of the MP/adepithelial cells retain a more normal morphology. In the relatively mild conditions from ∆Enh3 or from *miR-8* overexpression, there is residual Zfh1 present and most of the cells retain their normal morphology rather than becoming differentiated.

Thus the differences resemble those commonly seen with alleles of varying severity (so-called “allelic series”) as expected if the experiments result in different degrees of *zfh1* depletion.

3) –In the subsection “An alternate short zfh1 isoform is transcribed in adult MPs” and elsewhere the authors argue repeatedly that Zfh1 expression and thus stem cell maintenance must "evade" regulation by miR-8, which they explain by the expression of the short isoform that lacks miR-8 target sequences. However, in Figure 4 they show that mir-8 is not even expressed in the satellite-like cells, which means there is actually no need for evasion in these cells. Currently, there is only evidence that miR-8 is expressed in fully differentiated muscles, which could be downstream of differentiation regulators such as Mef2 (in line with data from Figure 6—figure supplement 1). So the proposed evasion mechanism is highly speculative and cannot be presented as a fact without any supporting data. Such data could for instance come from G-TRACE experiments with miR-8-GAL4 that show that miR-8 is indeed transiently expressed in the precursors of the satellite-like cells.

We agree that the “evasion” model implies that *mIR-8* expression should overlap with Zfh1 during the stages where pMPs are selected and we appreciate the reviewer’s suggestions. To address this point we have examined the expression of *mIR-8* and Zfh1 during pupal stages (at circa 20hr and 30hr after pupa formation) when the pMPs are specified. These data, which are shown in new Figure 5D-F(subsection “*zfh1* is silenced by the conserved microRNA *miR-8/miR-200* in MPs”, third paragraph) demonstrate that there is a phase where the two overlap in their expression. At 20hr the differentiating myoblasts have higher levels of *mIR-8* than the remaining MPs while the converse is seen for Zfh-1, which is at highest levels in the MPs. By 30 hours, few Zfh1 expressing cells remain and a small subset of these retain *miR-8* expression (new Figure 5D-F). Together these new data show that *miR-8* is initially expressed in the precursors and that there is an important window where its expression overlaps with that of Zfh1 in the pMPs

4) Subsection “zfh1 isoform transcription requires Notch activity in adult SCs”, second paragraph, Figure 6, Figure 8—figure supplement 1: The authors ascribe the high Mef2 levels (i.e., possible entry into differentiation) in satellite cells with depleted Notch to reduced zfh1-short. However, as they show, zfh1-long is unaffected by Notch depletion and because mir-8 is absent in these cells (see above), zfh1 could still suppress differentiation. This would argue for zfh1-independent effects of Notch.

The reviewer is correct that our data argue that the main effect of perturbing Notch will be on *zfh1short* and that residual *zfh1-long* in the pMPs would, in principle, be able to prevent differentiation. However the levels of *zfh1-long* at this stage are comparatively low, so it is likely that the amounts of Zfh1 in the Notch-depleted pMPs are not sufficiently high to robustly inhibit the differentiation in all of the pMPs. We have modified the text (Discussion) to explain these points and included the caveat that there could also be *zfh1* independent effects of Notch as highlighted by the reviewer.

Reviewer #2:[…] I have some critics and specific comments that need clarification:1) Role of Zfh1 short RNA isoform in the maintenance of muscle progenitors. Some additional experiments, such as gain-of-expression of zfh1 isoforms, specifically the short form to substantiate the model.

To show that Zfh1-short can counteract muscle differentiation, we have generated a UAS-Zfh1Short transgenic line and shown that its expression can prevent the premature differentiation phenotype caused by over-expressing *Mef2* in the muscle progenitors. These new data, which have been added to new Figure 7—figure supplement 1 (subsection “An alternate short *zfh1* isoform is transcribed in adult pMPs”, last paragraph), show that Zfh1-short protein isoform retains the ability to antagonize *Mef2*, as shown originally for Zfh1-long (Siles et al.,2013), and support the model that Zfh1-short protects the MPs from differentiation.

2) Loss of mir-8 assays is paramount to the model. The mir-8 sponge should produce phenotypes that substantiate the biological relevance of the regulatory switch to maintain a subset of MPs retaining zfh1. mir-8 loss would be expected to increase the number of pMPs.

We thank the reviewer for this helpful suggestion. We have now tested the consequences from expressing the *miR-8* sponge specifically in the muscle progenitors during the differentiation phase, (using *Mef2*-Gal4). As predicted by the reviewer, this “mopping up” of *miR-8* led to a significant increase in the number of adult pMPs. These new data, which have been added to new Figure 6 and discussed in the main text (subsection “*zfh1* is silenced by the conserved microRNA *miR-8/miR-200* in MPs”, last two paragraphs), further substantiate the model that *mIR-8* counteracts the maintenance of pMPs.

3) Epistatic studies to link mir-8 overexpression to zfh1 (gain of mir-8 along with gain of zfh1 and/or loss of mir-8 and loss of zfh1). These experiments seem important for substantiating the biological relevance of the regulatory loci discovered.

Because *mIR-8* and Zfh1 are ultimately required in complementary cells, it is difficult to design ideal experiments combining their misexpression/knockdown along the lines proposed by the reviewer, especially because over-expression of Zfh1-long causes lethality. We have however performed several additional experiments to test the different aspects of the model individually. For example, we have over-expressed *miR-8* in the pMPs, where it leads to a reduction in pMPs consistent with *miR-8* shifting the balance towards differentiation (Figure 6—figure supplement 1 and subsection “*zfh1* is silenced by the conserved microRNA *miR-8/miR-200* in MPs”, last two paragraphs). We have expressed the *mIR-8* sponge, as summarized above, to deplete *mIR-8* and shown that this increases the number of MPs as predicted (new Figure 6, see the aforementioned paragraphs). We have demonstrated that *zfh1-short* expression can prevent muscle differentiation induced by *Mef2* (new Figure 7—figure supplement 1; subsection “An alternate short *zfh1* isoform is transcribed in adult pMPs”, last paragraph). We believe that these additional experiments greatly substantiate the regulatory logic that we propose.

4) The conclusion that adult Zfh1+ve cells retain capacity to divide and produce progeny should be reinforced with direct data. G-TRACE is not a cell lineage method – it traces the dynamics of a Gal4 expressing cell. It is very likely the authors' conclusion is right but a clonal cell lineage method is more appropriate for this particular issue.

We appreciate the concerns of the reviewer, which were also raised by reviewer 1 (point 1). Due to the nature of the tissue, it is very difficult to perform a classic clonal lineage tracing (we have tried extensively without real success). We have therefore taken the strategy suggested by reviewer 1, namely to induce G-TRACE expression only in the adult by restricting it with Gal80ts until after eclosion. When we use Gal80ts to conditionally activate the Enh3-Gal4 in adult flies and trace the progeny with G-Trace, we detect a limited number of GFP-positive nuclei present within adult muscles 10 days after they had emerged (new Figure 3C-D, subsection “Adult Zfh1+ve MP cells contribute to flight muscles”, last paragraph). These GFP positive nuclei were most concentrated in proximity to the satellite-like cells supporting the model that they are derived from the MPs. Furthermore, in our initial experiments we focused on the perdurance of the RFP signal in the newly born myoblasts to link them directly to the cells of origin (Figure 3A, subsection “Adult Zfh1+ve MP cells contribute to flight muscles”, first paragraph), because of the caveats mentioned by the reviewer. Finally we have performed new experiments ablating the pMP cells and have shown that this reduces the number of muscle nuclei (new Figure 3I, subsection “Adult Zfh1+ve MP cells contribute to flight muscles”, last paragraph), supporting the conclusion that the pMPs give rise to progeny that contribute to the muscles. We believe that together these data provide clear evidence that the pMPs retain capacity to divide and produce progeny that contribute to the adult muscles.

In summary, this is a very interesting study with a conceptually novel regulatory logic that may have broader implications for understanding how other stem cells and cancer stem-like cells may escape terminal differentiation. Nonetheless, some additional data would be required to substantiate that the zfh1 short isoform contributes (or is capable) to the maintenance of satellite-like progenitor cells.

We are glad that the reviewer finds the work conceptually novel and important. We believe that the additional data we have provided in the revised version substantiate our model that Zfh1-short contributes to progenitor maintenance.

Reviewer #3:In the submitted manuscript H. Boukhatmi and S. Bray analyse an intriguing issue related to the mechanisms of maintenance of persistent Muscle Progenitors (pMPs) in adult Drosophila muscles focusing on flight muscles. Considering that identification and functional characterisation of Drosophila muscle satellite-like cells that express Zfh-1 has already been reported (open access BioRxiv Chaturvedi et al., manuscript posted 25 Jan 2016 from VijayRaghavan lab) the Boukhatmi and Bray's manuscript carries two new findings: i) pMPs preferentially express a short isoform of zfh1, which is devoid of miR8 seed site and ii) zfh1 is not only present in pMPs but also plays an important role in their maintenance. Authors also identify and apply a new tool, zfh1-Enh3-Gal4 that allows them to target/visualize pMPs.My comments focus on these two aspects, which make this manuscript of potential interest for eLife if documented more rigorously.

We are glad the reviewer considers that our results are of potential interest for *eLife*. We would like to emphasise the fact that our data address the mechanism of how muscle satellite cells are maintained and demonstrate their role in normal muscle homeostasis. As noted by the reviewer this is a substantial advance on the recent work from VijayRaghavan’s lab, which very nicely documented the existence of the persistent muscle precursor/satellite cell population and showed their expansion during muscle injury (Chaturvedi et al., 2017). The reviewer is correct that they also reported that Zfh-1 is expressed in these cells, although these data were only added to their manuscript during the final round of revisions, July 2017, at the same time ours was initially submitted. We have modified our text to take on board the additional features in their final published version. However we stress that they have not demonstrated that Zfh1 is functionally important nor shed any light on its regulation, which are key the aspects of our work and that lead to a novel model for stem cell maintenance.

1) Differential post-transcriptional regulation of zfh1 isoforms by miR8 could indeed represent a major regulatory mechanism in setting pMPs population. Authors have tools in their hands to document when this isoform switch takes place. In Figure 2D they show that in 3rd instar larvae Enh3>GFP is largely expressed in Zfh1/Cut positive MPs in the notum region and then they jump to adult stage to show Enh3-GFP restricted to pMPs only. It would be interesting to see when in development this restriction is effective. In other words when the satellite-like pMPs could be identified and visualized for the first time. Time lapse imaging of IFM development in Enh3-GFP pupa (approach developed in Schnorrer lab) could represent an interesting view point as well. The developmental restriction of Enh3-GFP expression should be correlated with miR8 sensor and/or mir8-Gal4 expression to evaluate whether these two regulatory events are coordinated. In relation to this it would be important to better correlate Enh3 activity with transcriptional regulation of zfh1-short isoform.

We agree with the reviewer that there was a gap in our analysis of Enh3-GFP and thank them for encouraging us to investigate the timing of its developmental restriction. Following their suggestions we have carried out two further sets of experiments. In the first, we have followed Enh3_GFP and Zfh1 expression at two stages during pupal development, 20h APF and 30h APF, see new Figure 2F, G and subsection “*zfh1* enhancers conferring expression in MPs”, third paragraph. Both Enh3_GFP and Zfh1 are broadly expressed in myoblasts and in MPs at 20h, where a subset of unfused MPs express higher levels of both. By 30hrs Enh3_GFP and Zfh1 are restricted to a few scattered cells, the presumptive pMPs, which are closely apposed to the muscle fibers. At the same stage, a small number of Zfh1+ve cells lack detectable Enh3 expression and we speculate that these are undergoing differentiation. These data suggest that expression from Enh3 correlates with the restriction/selection of the pMPs, which begins around 20 hrs and is largely completed by 30 hrs.

In the second experiments we have investigated *mIR-8* expression at similar stages (new Figure 5, subsection “*zfh1* is silenced by the conserved microRNA *miR-8/miR-200* in MPs”). At 18-22hr APF *miR-8-Gal4* and Zfh1 expression overlap in most if not all muscle nuclei (new Figure 5D). However *miR-8-Gal4* expression level was elevated in the differentiated myoblasts, whereas Zfh1 expression level was slightly lower in this population and higher in the undifferentiated MPs. By 30h APF, their expression becomes mutually exclusive, except in a few rare MPs where both are co-expressed (new Figure 5E). Thus *mIR-8* and Zfh1 develop reciprocal expression profiles between 20 and 30hr APF, as predicted if *mIR-8* down-regulates Zfh1. The timing of this restriction is similar to that seen for Enh3, supporting the model that Enh3 is responsible for the persistence of Zfh1 in MPs.

It is unclear whether Enh3 indeed regulates preferentially transcription of zfh1-short. Levels of short and long transcripts could be quantified in ΔdeltaEnh3 context in wing discs (instead of Zfh1 protein shown in Figure 2G) and in adult pMPs.

We agree with the reviewer that we have not fully ruled out the possibility that Enh3 could also regulate *zfh-1-long*. We are hampered by the fact that it is difficult to quantify the transcript levels unless we can reliably measure the individual foci in the FISH, so we could not reliably assess changes in *zfh-1-long* in the wing disc in ∆Enh3. In addition the early lethality of ∆Enh3 means that it is not plausible to monitor the isoforms in the adult pMPs. What we have shown is that Enh3 is the only tested *zfh1* enhancer that is activated and maintained in the pMPs, where *zfh1-short* is the predominant isoform (Figure 2 and 7) and that depletion or up-regulation of Notch, which acts through Enh3, specifically affects *zfh1-short* expression (Figure 8). We cannot exclude that Enh3 also regulates *zfh1-long* in some circumstances, especially as the region defined is 3kb and may encompass more than one regulatory element. We have commented specifically on the possibility in the text (subsection “*zfh1*-short isoform transcription requires Notch activity in adult pMPs” and Figure legend 9).

2) Regarding second major finding that zfh1 is required for maintenance of pMPs it could be important to test functional relevance of the two isoforms. Enh3>miR8 could be applied to deplete pMPs of zfh1-long and Enh3>NRNAi to see if effects of N and zfh1RNAi are similar (with experimental setup like in Figure 7). Generation of CRISPRs mutation of zfh1-short promoter/TSS site could help in defining function of this isoform.

We have taken on board the thoughtful suggestions of the reviewer to explore further the role of Zfh1 short by comparing the effects from expressing N-RNAi and *zfh1*-RNAi in the adult pMPs using Gal80ts in combination with Enh3-Gal4 so the knock down occurs only in the adults (new Figure 8I, subsection “*zfh1*-short isoform transcription requires Notch activity in adult pMPs”, last paragraph). Both have similar effects on the muscles, reducing the number of nuclei in the same way as when the pMPs are ablated genetically (new Figure 3). As Notch primarily promotes expression of the *zfh1-short*, this supports the model that this isoform is important in preventing MPs from differentiating. While the generation of a CRISPR mutation specific for *zfh1-short* is potentially a powerful way to further test the role of this isoform, unfortunately it is lethal at an early stage (preliminary unpublished data from a colleague), due to expression of this isoform in embryos, so it is not possible to analyze the adult muscle phenotype.

As suggested by the reviewer, we have also now tested the consequences of forced *miR-8* expression in the adult (new Figure 6—figure supplement 1D) and found that it causes a reduction in pMPs. The fact that the pMPs remain sensitive to miR-8, which would only target *zfh1-long*, suggests that the levels of Zfh1 are finely tuned by the expression of both *zfh1-long* and *zfh1-short* in these cells. We have explicitly discussed this possibility in the Discussion.

3) Referring to the role of pMPs in muscle homeostasis one complementary experiment could be to test effects of ablation of these cells in adult flies (Enh3>Rpr).

We thank the reviewer for this thoughtful suggestion. We have, as they proposed, performed a genetic ablation of the pMPs in adult MPs by expressing the proapoptotic gene *reaper*. As predicted this has led to a decrease in muscle nuclei number (new Figure 3, subsection “Adult Zfh1+ve MP cells contribute to flight muscles”, last paragraph). These results are consistent with those from *zfh1* RNAi assays and provide a complementary experiment showing that the pMPs have an important role in muscle homeostasis.

[Editors' note: the author responses to the re-review follow.]

Reviewer #2:[…] The images they provide are convincing but regarding the intensity it would be good to have further information in how they had performed the analysis. Usually, it is a good idea to do flip-out clones so that there is internal control and one can appreciate the different intensity between the mutant cells and the wt surrounding within the same staining. However this might be difficult for some experiment here.

We are glad the reviewer is positive about all the additional data we have included. We appreciate their concerns about analysis of staining intensities and agree that in tissues where flip-out clones are effective this is a powerful way to have internal controls. Unfortunately, because the cells become dispersed, the method does not work well in this tissue. However, in all cases the experiments have been replicated and we now present the data from those replicates by plotting all of the data points and shading them accordingly. We hope that the more transparent and informative plotting of all our data points demonstrates clearly the robustness of our replicate data sets and addresses these concerns.